


# 3-D tomographic observations of Rossby wave breaking over the Northern Atlantic during the WISE aircraft campaign in 2017

Lukas Krasauskas[1], Jörn Ungermann[1,2], Peter Preusse[1], Felix Friedl-Vallon[3], Andreas Zahn[3], Helmut Ziereis[4], Christian Rolf[1], Felix Plöger[1], Paul Konopka[1], Bärbel Vogel[1], and Martin Riese[1]

[1]Institute of Energy and Climate Research – Stratosphere (IEK-7), Forschungszentrum Jülich, Germany
[2]Jülich Aachen Research Alliance (JARA), Forschungszentrum Jülich, Germany
[3]The Institute of Meteorology and Climate Research (IMK), Karlsruhe Institute of Technology, Germany
[4]The German Aerospace Center (Deutsches Zentrum für Luft- und Raumfahrt; DLR), Institute for Atmospheric Physics

**Correspondence:** L. Krasauskas (l.krasauskas@fz-juelich.de)

**Abstract.** This paper presents measurements of ozone, water vapour and nitric acid ($HNO_3$) in the upper troposphere/lower stratosphere (UTLS) over North Atlantic and Europe. The measurements were acquired with the Gimballed Limb Observer for Radiance Imaging of the Atmosphere (GLORIA) during the Wave Driven Isentropic Exchange (WISE) campaign in October 2017. GLORIA is an airborne limb imager capable of acquiring both 2-D data sets (curtains along the flight path) and, when

the carrier aircraft is flying around the observed air mass, spatially highly resolved 3-D tomographic data. Here we present a case study of a Rossby wave (RW) breaking event observed during two subsequent flights two days apart. RW breaking is known to steepen tracer gradients and facilitate stratosphere-troposphere exchange (STE). Our measurements reveal complex spatial structures in stratospheric tracers (ozone and nitric acid) with multiple vertically stacked filaments. Backward trajectory analysis is used to demonstrate that these features are related to several previous Rossby wave breaking events and that the

small-scale structure of the UTLS in the Rossby wave breaking region, which is otherwise very hard to observe, can be understood as stirring and mixing of air masses of tropospheric and stratospheric origin. It is also shown that a strong nitric acid enhancement observed just above the tropopause is likely a result of $NO_x$ production by lightning activity. The measurements showed signatures of enhanced mixing between stratospheric and tropospheric air near the polar jet with some transport of water vapour into the stratosphere. Some of the air masses seen in 3-D data were encountered again two days later, stretched to

very thin filament (horizontal thickness down to 30 km at some altitudes) rich in stratospheric tracers. This repeated measurement allowed us to directly observe and analyse the progress of mixing processes in a thin filament over two days. Our results provide direct insight into small-scale dynamics of the UTLS in the Rossby wave breaking region, witch is of great importance to understanding STE and poleward transport in the UTLS.

## 1 Introduction

The upper troposphere and lower stratosphere (UTLS) is the atmospheric region heavily influenced by both stratospheric and tropospheric air masses and is characterised by sharp gradients of many tracer concentrations in both vertical and horizontal



directions. The composition of UTLS plays a key role in stratosphere-troposphere exchange (STE) (Gettelman et al., 2011), has a large effect on radiative forcing (Forster and Shine, 1997; Riese et al., 2012) and hence on climate change.

Rossby wave activity (e. g. Salby, 1984) is one of the major processes that shapes the dynamics and structure of UTLS.
In particular, Rossby wave (RW) breaking is known to steepen the tracer gradients and act as a horizontal transport barrier (McIntyre and Palmer, 1983, 1984) and facilitate STE (Waugh et al., 1994; Günther et al., 2008; Rolf et al., 2015). RW breaking events can transport wet polluted air masses from troposphere into the lower stratosphere, which is important for the transport of ozone-depleting very short-lived substances and radiatively active substances into the lower stratosphere. The tendency of RW breaking events to create complex spatial distributions of tracers, such as very thin filamentary structures, has been known for
some time both from theoretical work (Polvani and Plumb, 1992; Scott and Cammas, 2002) and studies combining modelling with *in situ* aircraft measurements (e.g. Waugh et al. 1994, Vogel et al. 2011, Konopka and Pan (2012)) or satellite data (Bacmeister et al., 1999; Pan et al., 2009). RW breaking plays an important role in the formation of the extratropical transition layer (ExTL; Shapiro, 1980; Kritz et al., 1991; Hoor et al., 2004), characterised by mixing of tropospheric and stratospheric air masses. Quantifying these mixing processes as well as modelling ExTL structure in general is still a major challenge
(e.g. Hegglin et al., 2010). Observing RW breaking, as well as the related mixing and transport, remains difficult: satellite instruments have global coverage, but typically lack resolution (vertical resolution in the UTLS is particularly problematic) for the characteristically small scales involved in wave breaking, while *in situ* observations from aircraft are very highly resolved and accurate, but their spatial coverage is limited to the flight path of the carrier aircraft. Aircraft based limb sounders provide the very attractive middle ground between satellites and *in situ* with better resolution than the former and much better spatial
coverage than the latter.

In this paper, we present data acquired by the Gimballed Limb Observer for Radiance Imaging of the Atmosphere (GLORIA; Friedl-Vallon et al., 2014), an airborne imaging IR spectrometer. It can be used to retrieve temperature and volume mixing ratios (VMR) of a number of tracers, including ozone, water vapour, nitric acid ($HNO_3$), trichlorofluoromethane (CFC-11), chlorine nitrate ($ClONO_2$), and peroxyacetyl nitrate (PAN) among others. Besides the 2-D slices of the atmosphere acquired
by combining a set of atmospheric profiles along the aircraft flight path, GLORIA can be flown around an air volume to acquire a data set suitable for 3-D tomographic retrieval of temperature and trace gas concentrations (Ungermann et al., 2010). 3-D data products have measurement resolution better than 0.2 km vertically and 20 km horizontally. Filamentation and mixing are inherently 3-D processes, their full analysis requires dimensions of features in all spatial directions. These are immediately available when working with 3-D data, while *in situ* data often involves complex theory or model based argumentation to
compensate for the sparse measurements. Before this study, GLORIA 3-D retrievals were used to study internal gravity waves (Krisch et al., 2017).

The main focus of this case study is to determine to what extent the small scale tracer structure of UTLS around the polar jet can be understood as a consequence of Rossby wave related stratosphere-troposphere exchange and how this structure is effected by mixing. The outcome of this investigation could further be used to constrain mixing in models as well as to gain
insights into STE transport pathways.





The Wave Driven Isentropic Exchange (WISE) measurement campaign took place in Shannon, Ireland in September-October 2017. A flight on 7 October (Figure 1a) studied an early stage of a Rossby Wave breaking event. Tomographic 3-D GLORIA observations of ozone and nitric acid were performed around a heavily meandered part of the polar jet, resulting in a unique data set with signatures of multiple vertically stacked filaments. The origins of the filaments were studied using backward
trajectories computed by the Chemical Lagrangian model of the Stratosphere (CLaMS; Pommrich et al., 2014). The same air mass was observed again during the subsequent flight on 9 October, in the late stage of the wave breaking event (Figure 1b). This repeated measurement was used to directly observe mixing in progress by a match of air parcels and comparing the degree of mixing via tracer-tracer correlations. It is also shown that a strong nitric acid enhancement observed just above the tropopause could not have resulted from the descent of stratospheric air masses and is likely a result of $NO_x$ production by
lightning activity.

The paper is organised as follows. Section 2 briefly describes the GLORIA instrument, data analysis techniques and introduces the reader to the meteorological situation during the two research flights discussed in this paper. Measurement results and analysis of the observed phenomena are presented in Section 3. The effects of Rossby wave breaking on UTLS in the region around the polar jet is discussed in Section 3.1 by comparing the vicinity if the jet to surrounding areas in terms of
filamentation and tracer correlations. Section 3.2 presents in depth analysis of the small scale structure observed using 3-D reconstruction within the hexagonal flight pattern of 7 October flight. Section 3.3 is dedicated to nitric acid-ozone correlation analysis and the origins of the nitric acid enhancement observed just above tropopause. Section 3.4 presents analysis of the thin filament of stratospheric tracer rich air from 9 October flight and mixing analysis based on air parcels observed during both flights. Finally, conclusions are given in Section 4. The appendices provide technical information about data retrieval methods,
errors and validation.

## 2   Measurement and analysis techniques

### 2.1   GLORIA instrument and retrieval methods

The Gimballed Limb Observer of Air Radiance in the Atmosphere (GLORIA) is an airborne IR limb imager. It is a Michelson interferometer and acquires spectra in the 770 to 1400 $cm^{-1}$ range (Riese et al., 2005; Friedl-Vallon et al., 2014). Spectral resolution depends on measurement mode. The best possible spectral sampling of 0.0625 $cm^{-1}$ is achieved in the so called
*chemistry mode* used for retrieving a large number of trace species with often low mixing ratios. One full interferogram is recorded in about 13 s in this mode. In *dynamics mode*, spectral sampling is reduced to 0.2 $cm^{-1}$, but the faster acquisition rate of $\approx 5$ s per interferogram allows for instrument panning, i.e. the observation direction with respect to aircraft heading alternates between 10 values in between $45°$ and $135°$. This is useful for observing dynamically complex structures from
different directions. 3-D tomography images are acquired in this mode.

The detector of the spectrometer is a 2-D array with 128 x 48 effective pixels and $4° \times 1.5°$ field of view in vertical and horizontal directions, respectively. In limb observation geometry, the instrument looks in a close to horizontal direction. Infrared radiation along any line of sight comes mostly from the lowest altitude point on that line of sight (called *tangent point*). That





way, a large range of altitudes can be recorded with the 2-D detector in a single interferogram. This setup also allows for very high vertical resolution of up to 200 m. Limb observers can, however, only provide detailed information on the air masses at and below flight altitude. The WISE measurement campaign discussed here was conducted with the German HALO research aircraft at flight altitudes up to 15 km. All retrieved data is limited by this altitude.

Best resolution is achieved when the aircraft is flying around the observed air mass in a close-to-circular flight pattern and also panning the instrument (dynamics mode). Due to practical considerations, the actual tomography-optimised flight paths are typically hexagonal and around 400 km in diameter (Ungermann et al., 2010).

Both 2-D and 3-D retrievals are performed by means of inverse modelling, using the Jülich Rapid Spectral Simulation Code Version 2 (JURASSIC2). The radiative transfer model (Hoffmann et al., 2008) (employed as the forward model) uses the emissivity growth approximation method (Weinreb and Neuendorffer, 1973; Gordley and Russell, 1981) and the Curtis-Godson approximation (Curtis, 1952; Godson, 1953). Levenberg-Marquardt algorithm (Marquardt, 1963) and conjugate gradients solver (Hestenes and Stiefel, 1952) are used for inverse modelling. For more information about 3-D tomography implementation refer to Ungermann et al. (2010, 2011) and Krasauskas et al. (2019). In general, 3-D tomographic retrievals are computationally expensive, typically requiring a few hundred of CPU core-hours to complete.

The tomographic retrieval combines data from measurements taken over a period of 1 h and 45 min. Some locations are observed from one direction in the beginning of this period, and then again from another angle near the end of it. If the measured parameters of the atmosphere change meaningfully (i.e. statistically significantly above noise level) during that time, these observations will no longer agree. This problem is solved by introducing advection compensation (Ungermann et al., 2011). The main idea of the approach is as follows. We choose a time $t_v$ for which the retrieval will be valid (16:50 UTC in this case, approximately the middle of the measurement time window). As it is the case with most instruments, GLORIA retrieval data is provided on a discrete grid. If a measurement was taken at a time $t_m$, one can relate all required atmospheric parameters (temperature, trace gas concentrations) at each grid point at time $t_m$ to the actual retrieved quantities (atmospheric parameters at time $t_v$) by advecting grid points at time $t_m$ to time $t_v$ and reinterpolating. In practice, this advection is not performed for every $t_m$, but only to a number of preset values and then temporal interpolation is used.

**Table 1.** Spectral windows for 3-D retrieval

| # | Spectral range, cm$^{-1}$ | # | Spectral range, cm$^{-1}$ |
|---|---|---|---|
| 1 | 791.0 - 793.0 | 6 | 980.0 - 984.2 |
| 2 | 863.0 - 866.0 | 7 | 992.6 - 997.4 |
| 3 | 892.6 - 896.2 | 8 | 1000.6 - 10006.2 |
| 4 | 900.0 - 903.0 | 9 | 1010.0 - 1014.2 |
| 5 | 956.8 - 962.4 | | |

The 3-D retrievals of ozone and nitric acid ($HNO_3$) shown in this paper were performed using radiances from the spectral windows given in Table 1. The a priori data for air temperature and pressure was taken from the European Centre for Medium-Range Weather Forecasts (ECMWF Dee et al., 2011) operational analyses. Whole Atmosphere Community Climate Model (WACCM; e.g. Garcia et al., 2007) data was chosen as a priori for ozone and $HNO_3$.





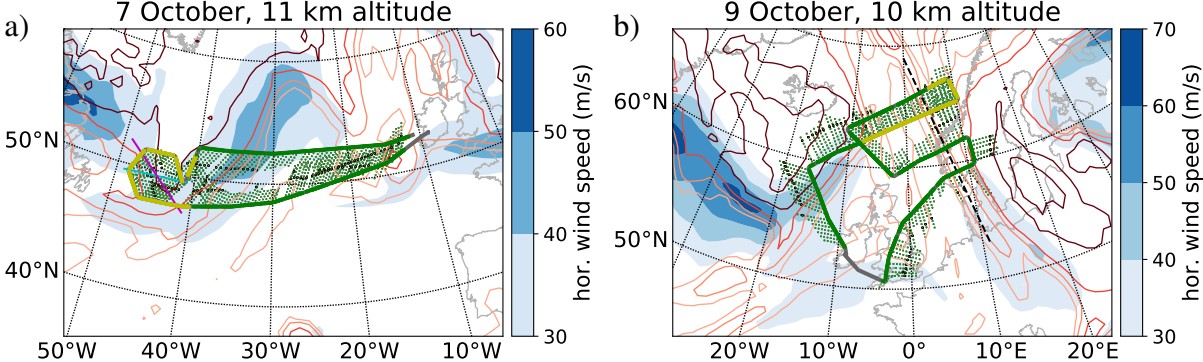

**Figure 1.** Synoptic situation during measurement flights. Flight track shown in green for chemistry mode, yellow for dynamics mode measurements. Tangent points are depicted as green dots. Potential vorticity contours for 2, 4, 6, and 8 PVU, respectively, are shown in increasingly darker reddish colours. Magenta and cyan lines in panel a) mark the position of cuts shown in Figure 7, dashed black line in panel b) marks a filament with older air from the stratosphere (see Section 2.2 for discussion)

## 2.2 Measurement flights

On 7 October 2017 an onset of a Rossby wave breaking event was predicted over the western part of North Atlantic using ECMWF forecast products. A research flight was planned to probe the UTLS above the polar jet (Figure 1a). The HALO
aircraft took off from Shannon, Ireland, at at around 12:30 UTC and landed just after 20:00 UTC. The GLORIA instrument performed well and measured for the whole duration of the flight. These measurements were used for 1-D retrievals, and the resulting profiles were combined into a 2-D data product (curtain) along the flight path.

A hexagonal flight pattern in the Rossby wave breaking region itself was flown between 16:00 UTC and 17:45 UTC, allowing for a 3-D tomographic analysis of the volume. Due to the limited aircraft range, flying a full hexagon this far from the operating
base was not possible, hence only 5 sides were executed. Also, after flying 2 of the 5 sides, HALO climbed from the altitude of 13.4 km to around 13.9 km. For best results and easier processing, one would wish to avoid climbing in the middle of the tomographic data acquisition. In this case, however, flying at the higher altitude was not possible earlier due to the weight of the remaining fuel. Also the benefits of flying the last 3 hexagon sides higher and hence observing a wider altitude range were deemed more important. This choice was further motivated by the presence of optically thick clouds at low altitudes. In parts
of the hexagon, cloud tops reached to almost 10.5 km altitude, thereby restricting the altitude range available for retrieval. Another complication for producing 3-D data was the high wind speeds of the polar jet, reaching more than 40 m/s, resulting in significant advection of air masses during the measurement acquisition time (see Section 2.1 for solution to this particular problem). The measured air masses could be successfully reconstructed in 3-D despite of these challenges. The tomographic retrieval is complemented by 2-D data (curtains of profiles obtained from 1D retrievals) of the region immediately east of the
hexagon, which also proved to be of interest.

By 9 October 2017, the wave breaking event was at its final stage and most of the air masses that were present in the hexagon flown two days before were stretched into a thin filament extending from 70°N, 0°E to 51°N, 12°E. This filament can be seen





as an elongated 2 PVU contour in Figure 1b and is highlighted by a dashed black line. A research flight on 9 October was flown to investigate the late stage of this RW breaking by measuring this filament.The aircraft crossed the filament a total of four times during this flight, as seen in Figure 1b. The flight lasted from 08:30 UTC till 17:00 UTC. In this paper, we present only 2-D data products from this flight.

## 2.3   Backward trajectories

Backward trajectory calculations from the chemical Lagrangian model of the Stratosphere (CLaMS; Pommrich et al. 2014) were used for measurement data analysis. CLaMS back trajectory calculations are very well suited to analyse the detailed transport pathway of an air parcel in the UTLS and were applied to a variety of problems such as transport in the tropics as well as STE processes (e.g. Ploeger et al. 2012, Vogel et al. 2014, Li et al. 2018). The model was driven by winds obtained from ECMWF operational analysis data. An air parcel was placed at each grid point of the GLORIA data product, and the trajectories of these parcels were traced back for up to 1 month. Mixing and chemical processes, which are normally included in the 3-D version of CLaMS model, are not used for backward trajectory calculations. One of the main uses of backward trajectories was to determine the "age-of-air", i.e. time that an air parcel in question spent in stratosphere. The maximum potential vorticity gradient tropopause (Kunz et al., 2011) was used to define stratospheric and tropospheric regions along the calculated backward trajectories. This tropopause definition is based on the product of isentropic PV gradients and wind speed. It is well-suited for our study as it is generally in good agreement with chemical discontinuity in trace gas distributions.

## 3   Results and discussion

### 3.1   Filamentation and mixing near and away from the polar jet

The results presented in this section are based on the GLORIA 2-D data products. They were generated along most parts of both flights considered in this work (Figure 1). The measured volume mixing ratios of ozone, water vapour and nitric acid for 7 October flight are presented in Figure 2. The retrieved values are shown as functions of time and altitude, but it is important to remember that the GLORIA 2-D curtains do not represent a vertical cut through the atmosphere along the flight path but a slant curtain. At flight altitude, the data shown represents the atmosphere $\sim 10$ km to the right of the aircraft, while at lowest altitude this distance increases to up to 250 km. The locations of the tangent points of the observations are shown in Figure 1a. The missing data in Figure 2 is due to aircraft manoeuvres and optically thick clouds. In the case of cloud presence, only the data above cloud top is available. The retrieval of 2-D data was also performed for the hexagonal part of the flight path (from 16:15 to 17:42 UTC) for consistency, even though the 3-D data is available for that part of the flight. 2-D data products are valid at the measurement acquisition time, indicated in the plots.

Generally, water vapour has high volume mixing ratios in the troposphere and is much less abundant in the stratosphere due to freezeout in particular around the tropical tropopause (Schiller et al., 2009). $O_3$ and $HNO_3$ are more abundant in stratosphere, where they are generated by photolysis, and less abundant in the troposphere. Generally these two gases have



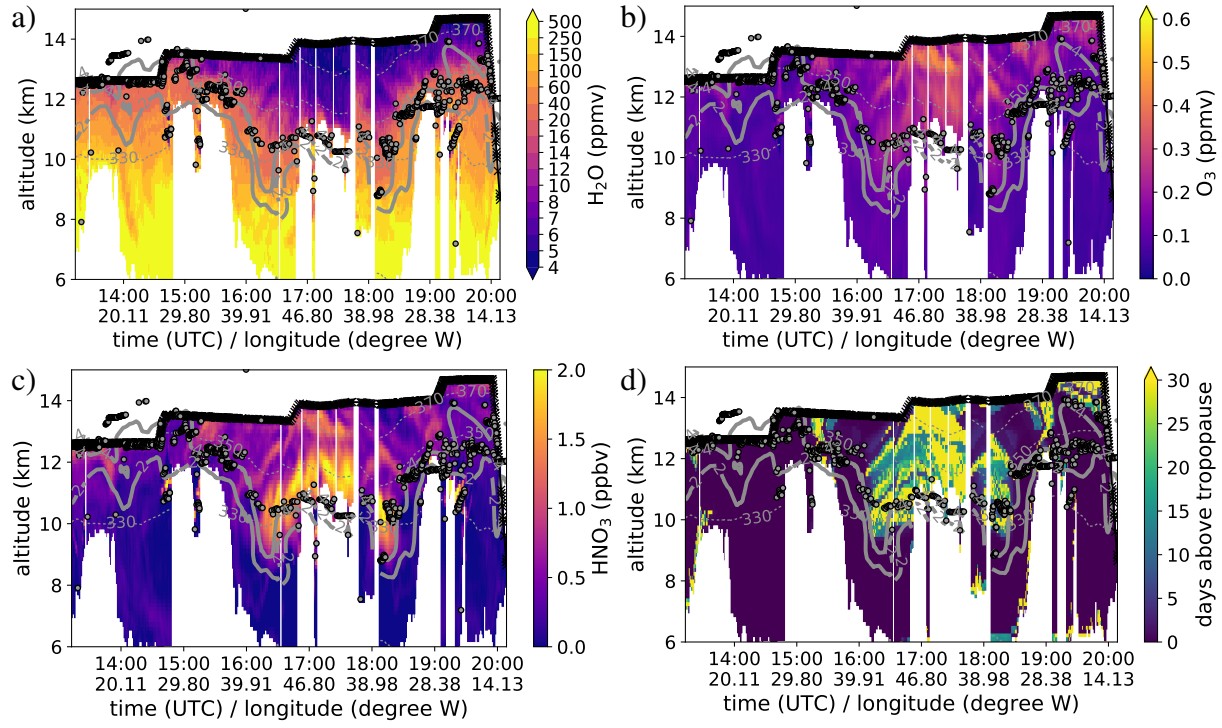

**Figure 2.** Panels a-c): 7 October $H_2O$, $O_3$ and $HNO_3$ 2-D GLORIA data along the flight path shown in Figure 1a. Panel d): air mass origin (details in Section 3.1). Isentropes are shown as dashed lines, 2 and 4 PVU contours – thick solid lines, thermal tropopause – dots.

a compact positive correlation in the deep stratosphere. The correlation is attributed mostly to source and sink regions being

similar for the two gases, and not a result of chemical interactions between them (Section 3.3; Popp et al., 2009). Retrievals

generally confirm the aforementioned trends, but also show heavily filamented structures in all the tracers presented here. There

is a clear positive correlation between ozone and nitric acid: ozone-rich air masses tend to have high nitric acid concentrations

and vice versa. One can also see negative correlation between water vapour and ozone (or, alternatively, between water vapour

and nitric acid), especially around 16:15 and 18:30 UTC, where the $H_2O$ structure shows more filamentation. This suggests

that the whole structure was formed by STE: air parcels rich in ozone and nitric acid (old air) come from higher up in the

stratosphere, while wet air masses have relatively recently resided in the troposphere (young air).

The origins of the filaments were investigated using CLaMS backward trajectory calculations. The typical mixdown time

scale (i.e. time needed for the mixing to erase all structure) for the Rossby wave surf zone in the lower stratosphere is of the

order of one month (Juckes and McIntyre, 1987). Hence, in this region, one may expect small scale ozone-poor structures to

be a consequence of transport of ozone-poor tropospheric air across the dynamical tropopause within the last month. Based

on this reasoning, we plotted the "age" (i.e., time spent above the maximum potential vorticity gradient tropopause[1]) of each

stratospheric air parcel observed by GLORIA (Figure 2d). There is good agreement between young air masses in Figure 2d and

---

[1]Note that this definition of age is different from the classical one, which is the elapsed time since an air parcel entered the stratosphere in the region of tropical tropopause in particular. Air masses that have just crossed the extratropical tropopause are considered young with our definition here.

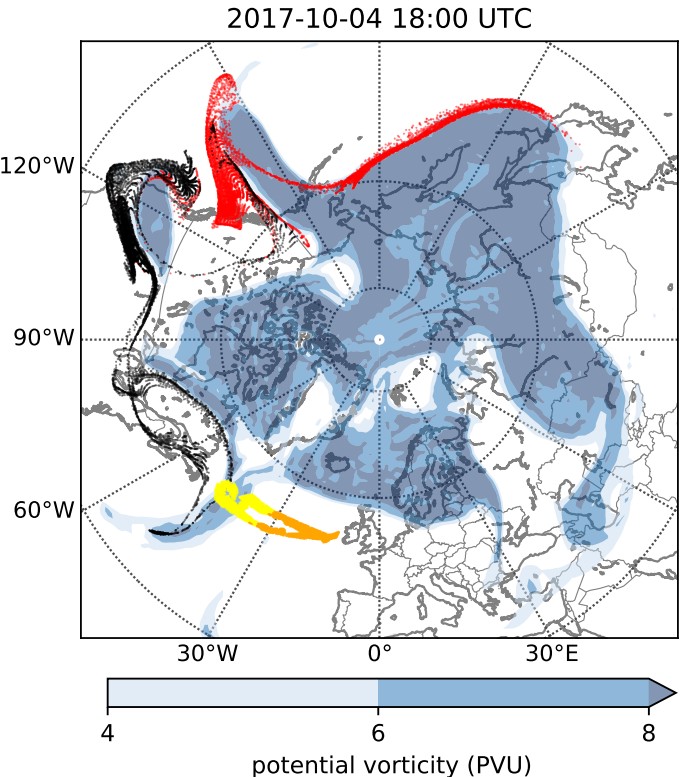

**Figure 3.** Air mass origin plot for the retrieved 2-D data. Horizontal cut at 11 km altitude 3 days before measurement flight is shown. Locations of GLORIA observations on 7 October: yellow - near polar jet (west from 30° W, PV > 1 PVU), orange – away from jet (all the rest). The positions of "yellow" air parcels 3 days before measurement are shown in red, of "orange" air parcels – in black.

ozone-poor and $HNO_3$-poor air in GLORIA retrieval (Figure 2b,c). Almost all ozone rich (>0.25 ppmv) air masses, including thin filaments, have stayed in the stratosphere for more than one month. Younger air, conversely, is almost always ozone poor. Also, wet air filaments correspond to young air masses from the troposphere, that can be seen best around 16:15 UTC, 10-13 km altitude; 17:50 UTC, 11.5 km altitude and 18:40 UTC, 13-14 km altitude. Unfortunately, the air mass within the hexagonal flight pattern (16:15 to 17:42 UTC) was very dry, significant water vapour structures could only be found around the very edges of the hexagon and were advected outwards while this pattern was flown. The 3-D tomographic water vapour retrieval was therefore not performed.

HALO took off from Ireland, which was, at the time of the flight (7 October), south of the polar jet and therefore not directly effected by air masses of the polar UTLS, the polar jet and Rossby waves that shape it. One can clearly see the jet (in wind speed) and Rossby waves (in PV) in Figure 1a. Conversely, the western part of the flight, including the hexagonal flight pattern, is directly on top of the polar jet and therefore heavily influenced by it and the RW surf zone. To illustrate this difference, we divided the observed air masses into those "near polar jet" (we defined these as air masses observed at locations west of 30°W and with PV > 1 PVU at 11 km altitude), and "away from the jet" (all the rest). The locations of observation for these air





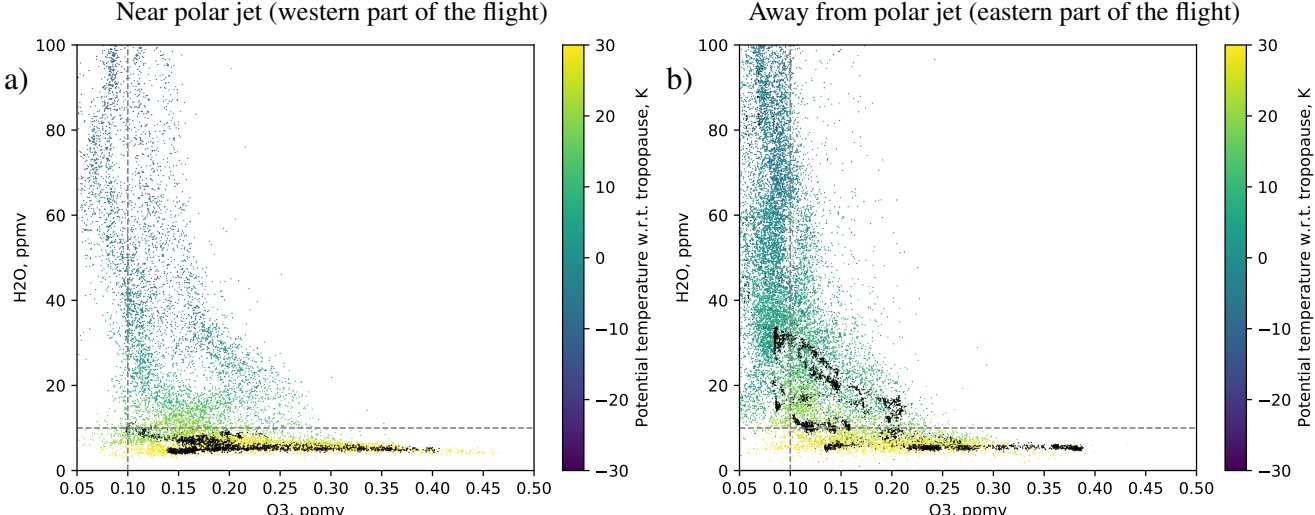

**Figure 4.** Ozone – water vapour tracer-tracer correlations from 7 October flight. Black dots represent *in situ* measurements at flight altitude. $O_3$ was measured by FAIRO (Zahn et al., 2012), $H_2O$ – by FISH (Meyer et al., 2015). GLORIA measurements are colour-coded by the difference in potential temperature relative to the tropopause.

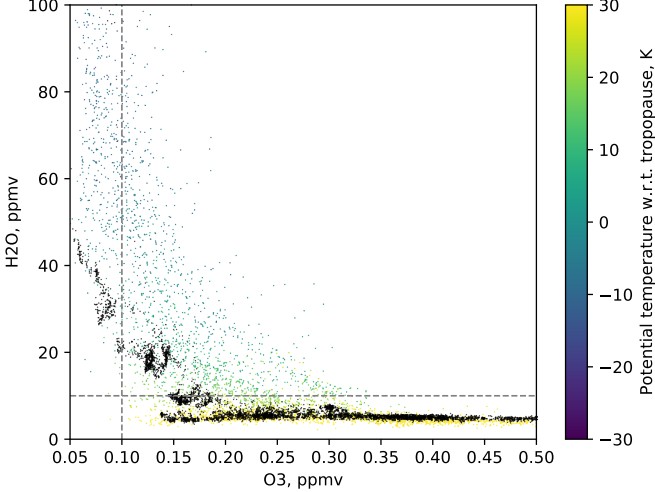

**Figure 5.** Ozone - water vapour tracer-tracer correlation from October 9 flight. Black dots represent in situ measurement data ($O_3$ – FAIRO, $H_2O$ – FISH), GLORIA measurements are colour-coded by the difference in potential temperature relative to the tropopause.

masses are shown in yellow and orange, respectively, in Figure 3. The PV at 11 km is plotted there as well and shows Rossby waves. The polar jet is located on the edges of the high-PV area (see Figure 1). The backward trajectory calculations show the position of the observed air parcels three days before observation. We see that the air masses "away from the jet" (black) came from lower latitudes and did not travel as far around the pole. The air found near the jet (red) has stayed near the jet for the last 3 days at least. The air parcels measured on 7 October as a rather compact group were stretched through a long portion of polar




jet on the other side of the pole just 3 days ago. These air masses were brought together by strong wind shear and transported far away by the strong westerlies of the jet. Note as well that the original positions of "near jet" air look especially dispersed around a breaking Rossby wave at 140° W. Such a sparse group of air parcels could only be brought into a compact group by strong wind shear and stirring in the breaking RW. This difference in air mass origin explains, why most of the filamentation

in trace gas structure can be seen between 15:10 and 19:10 UTC ("near jet" air). The origins of air inside the hexagonal flight pattern will be studied in more detail in Section 3.2.

To highlight the effect of RW breaking, we provide separate $O_3 - H_2O$ tracer-tracer plots for "near jet" air masses (Figure 4a) and the rest (Figure 4b). We designate air parcels as stratospheric, if they contain less than 10 ppm water vapour, and tropospheric, if they have less than 0.1 ppm ozone. Air parcels that fall into neither of these two categories are typically products of

mixing between stratospheric and tropospheric air (Proffitt et al., 1990; Hoor et al., 2002; Pan et al., 2004). When compared to in situ measurements, GLORIA data has lower spatial resolution, making it more difficult to identify individual mixing lines, but it has better spatial coverage allowing us to see a more complete picture of mixing across a range of altitudes. In the eastern part of the flight there is evidence of some mixing, but only between end members with relatively low values of both ozone and water vapour. This indicates that stratospheric air masses do not penetrate far into the troposphere and vice versa. Since

in situ observations represent a more spatially limited data set (black dots), two separate mixing lines can be discerned there. The "western" part of the flight, close to the polar jet, shows evidence of more vigorous mixing. In particular, there is a distinct branch of wet (> 20 ppmv water) air with potential temperatures consistent with the upper troposphere in this region and enhanced ozone (> 0.15 ppmv). This suggests an influx of stratospheric air into the upper troposphere. Also, the air mass with around 0.2 ppmv $O_3$ and 30 ppmv $H_2O$ could only have originated from a substantially deeper tropospheric intrusion into the

stratosphere (or vice versa) than that needed to explain the ozone-water vapour structure of the eastern part of the flight. As the aircraft stayed in stratosphere for the whole western part of the flight, all in situ data shows stratospheric signatures.

The $O_3$ - $H_2O$ correlation for the 9 October flight (Figure 5) has a similar dual mixing line structure as the corresponding correlation for "near jet" air parcels of the 7 October flight (Figure 4a), due to water vapour uplift into the stratosphere. This is not surprising and supports the previous result, since many of the air masses seen during the 9 October flight come from the

area observed on 7 October around the hexagonal flight pattern. The relative lack of purely tropospheric air masses is due to high clouds that prevented measurements in the troposphere (see Section 3.4 and Figure 12).

The combinations of ozone and water vapour concentrations observed by GLORIA during this flight are generally compatible with in situ observations during previous campaigns. For example, the air parcels indicating intense mixing, like the ones with 0.27 ppmv $O_3$, 20 ppmv $H_2O$ or 0.2 ppmv $O_3$, 50 ppmv $H_2O$ are within the range of values observed near Fairbanks,

Alaska during the POLARIS campaign in 1997 (Pan et al., 2009).

## 3.2   3-D tomography of filamentation near the polar jet

In this section we study the origin of the air inside the hexagonal tomographic flight pattern on 7 October (see Figure 1a). The flight pattern is shown, together with a 3-D plot of ozone results, in Figure 6. The results of the tomographic retrieval for ozone and nitric acid ($HNO_3$) are presented in detail as vertical 2-D cuts through the measured volume in Figure 7. There is,





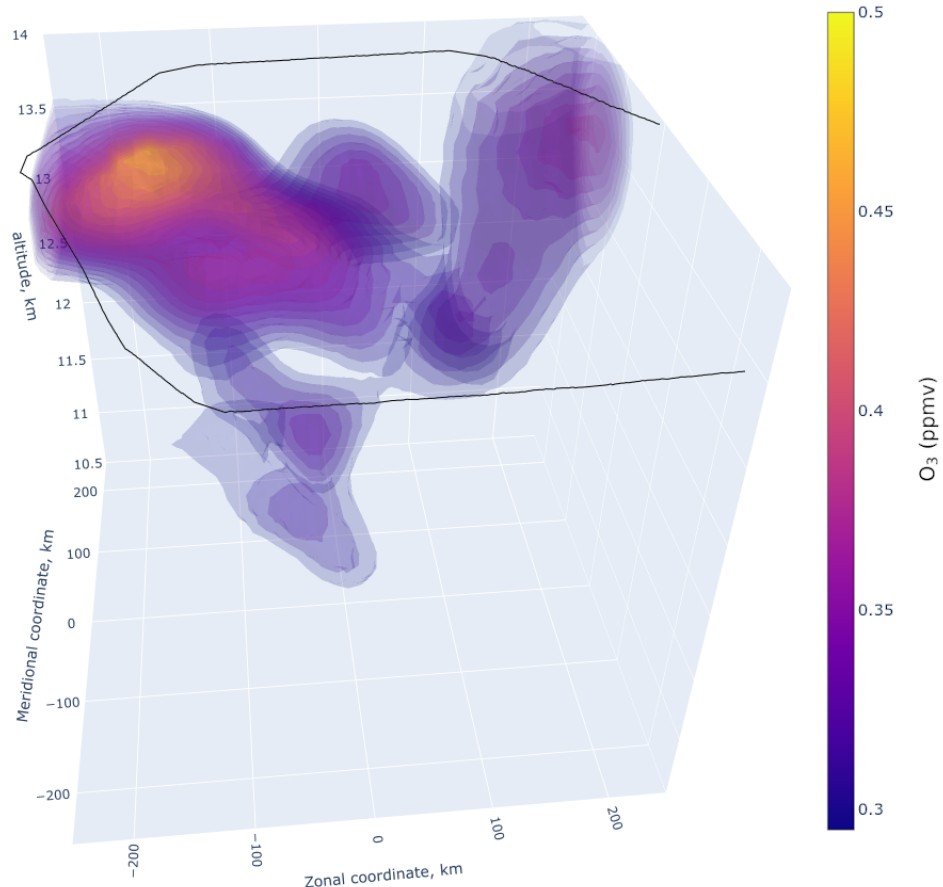

**Figure 6.** 3-D plot of tomographic retrieval of ozone VMR. Flight track represented by thin black line.

as before, a clear positive correlation between the two trace gases. A more detailed look into this correlation is presented in Section 3.3.

Figure 7 also shows a complex structure of thin filaments (layers) of younger and older air in the lower stratosphere (LS). As in the previous section, we use backward trajectories to determine the "age" (time spent above dynamical tropopause) of each stratospheric air parcel observed by GLORIA (Figure 7e, f). The two older air filaments (marked with white dots) that

are the main features of both ozone and nitric acid retrievals can be identified with the two filaments of air that has stayed in the stratosphere for more than 30 days. Younger air masses can be seen in between the two filaments as well as above them. The boundary between older stratospheric air and the troposphere at 10-11 km altitudes is also reflected in the retrieval. For ozone, in particular, values higher than approximately 0.25 ppmv within the hexagonal flight pattern are mostly measured for air parcels that stayed in the stratosphere for more than 30 days. The accuracy and resolution of the retrieval decreases away

from the centre of the hexagonal flight pattern, which is the likely reason for the "smearing out" of the filaments at around 48°N, 42.5°W.



**Figure 7.** Vertical cuts through 3-D tomographic retrieval and corresponding air mass origin plots. Solid grey line – projection of flight path onto the displayed cut, black dashed lines – isentropes, labelled in K, white dashed lines – PV isosurfaces, labelled in PVU. Black and red lines in the bottom panels show the position of air parcels for which backward trajectories were calculated. Their previous positions are shown, with those same colours, in Figure 8. Positions of vertical cuts are shown in Figure 1a: left panels - magenta, right panels - cyan. White dots mark filaments (see Section 3.2).



**Figure 8.** Origins of observed air parcels. Panels show horizontal cuts at 11 km altitude. Potential vorticity is shown in shades of blue. The location where air parcels were observed by means of GLORIA 3-D tomography on 7 October is shown in yellow (they form a small disc). The parcel locations at the dates indicated for each panel are shown in black and red. We further distinguish those into: red - those that were east of 165°W on 4 October 18:00 UTC, black – the remaining ones.

The panels of Figure 7 with GLORIA data also show potential temperature isolines, derived from GLORIA temperature retrieval and ECMWF pressure data. The major filaments are almost uniform in nitric acid and especially ozone structure, despite the fact that they are not isentropic. This suggests that diabatic processes or turbulence due to Rossby wave breaking were involved in their formation. The arrangement of tracer filaments and potential vorticity with respect to potential temperature surfaces shows that scale breakdown has occurred: although the filaments remain quasi-horizontal, they can subsequently be erased by mixing along isentropes, making the effective horizontal length scales shorter ($\approx 100$ km around the upper filament).

**Figure 9.** Panel a) shows the potential temperature of tropopause crossing for the parcels that crossed tropopause within the last month. Panel b) shows the change in potential temperature of observed air parcels within the last month. Black circles in panels a) and b) mark retrieved old air filaments, they are at the same positions as the white circles in Figure 7. Panel c) shows the distribution of potential temperature of tropopause crossing for the observed air parcels (5 K running mean).

A more detailed look into the backward trajectories allows us to trace the origin of each major filament and explain most of the observed ozone structure in terms of planetary wave activity. Figure 8a shows that air masses sampled by the 3-D retrieval on 7 October (location highlighted in yellow) come from an elongated region along the polar jet (edge of high PV region). This is because the retrieval sampled air masses right on top of the jet at different altitudes. Air parcels in the jet tend to stay confined to it, but are subject to extreme vertical wind shear: in the observed altitude range the wind speed of the jet decreases rapidly with altitude, thus air masses from low altitudes have travelled a longer distance around the pole. This can be confirmed in Figure 7e-f: the "red" air parcels of Figure 8, where observed below the "black" ones. A breaking Rossby wave can be seen





around 140°W on 4 October. The "black" group of air parcels were brought together by this breaking event: one can already see them as a relatively compact group on 4 October, but the panel from 29 September shows that they in fact come from very different locations west of the breaking wave. The old and young air structures seen in the "black" areas in Figure 7e-f are hence a consequence of this event: the young air masses (they are roughly 3 days old, as the event happened 3 days ago) entered the stratosphere because of the event, and older air masses were completely reshaped. In a similar fashion, another Rossby wave breaking event around 8 days prior to measurement (29 September) created the other prominent young air filament (seen at the top of the region marked by red contours in Figure 7e-f), with most of the air entering from the troposphere over western Europe. Most of the remaining air masses observed by GLORIA below 12 km can be traced back to a third wave breaking event on 25 September (Figure 8c). One can then see that the RW breaking is the dominant process for tracer structure genesis in this region, with most of the old versus young air structure formed by it. The 3-D data set could further be compared to models (such as CLaMS) with full mixing implementation enabled to evaluate the performance of such implementations and help to determine the model mixing parameters that would be appropriate for the extratropical UTLS, but this is outside the scope of this paper.

Figure 9 gives some insight into the vertical transport of observed air parcels. Panel c) shows the distribution of the observed air parcels according to the potential temperature at which they entered stratosphere (maximum potential vorticity gradient tropopause was used to determine the entry point). It shows the two distinct pathways of air into the stratosphere around the polar jet: from low potential temperature levels (∼340 K or less) upwards across the tropopause, or isentropically from lower latitudes at high potential temperature (horizontal transport). The latter pathway plays a major role, as expected for this region (Holton et al., 1995; Pan et al., 2009). Figure 9b shows the origin of air parcels in the vertical direction. We can distinguish between the two old air filaments that have descended from the stratosphere (black circles) that match the retrieved old air structures, the air in between and to the south east of the filaments, that has been isentropically transported into the stratosphere and has less stratospheric tracers, and the turbulent troposphere below ≈9.5 km with low ozone and nitric acid values. The young air filament just above the upper filament marked in circles has been strongly displaced upwards during a Rossby wave breaking event. Its signature can be seen in Figure 7a between the upper old air filament marked with white circles and a small old air blob at 43.5°W, around 13 km altitude.

## 3.3 Origins of nitric acid enhancement above the tropopause

Nitric acid ($HNO_3$) is produced in the stratosphere (mainly from $NO_x$) and removed in the troposphere by washout. Hence tropospheric $HNO_3$ mixing ratios can be as low as 0.1 ppbv or less. Due to similar source and sink regions, $HNO_3$ typically displays very compact relationship with ozone in the stratosphere. Popp et al. (2009) give a simple linear relation between the two gases, which typically holds within 15% in the stratosphere for ozone concentrations higher that 150 ppbv

$$HNO_3 = (0.00256 \pm 0.000154)O_3 - (0.0922 \pm 0.0886)\text{ppbv} \tag{1}$$





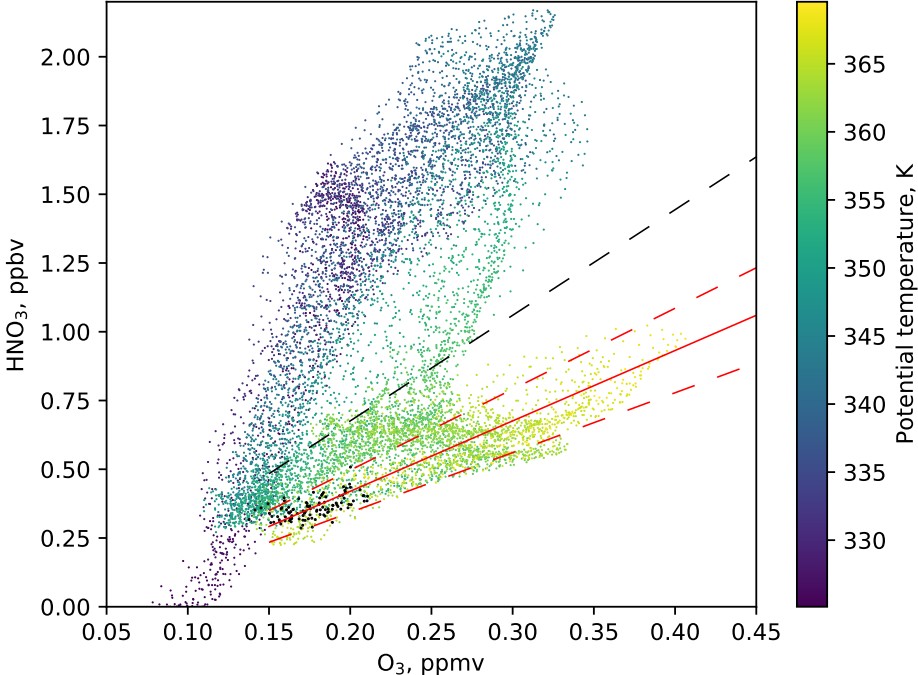

**Figure 10.** Ozone - nitric acid tracer-tracer correlation. Solid red line represents linear relationship of ozone and nitric acid typical for the stratosphere. Dashed red lines indicate the ±15% $HNO_3$ compared to that relationship, black dashed line indicates +50% $HNO_3$. Black dots represent in situ measurement data (see Appendix C for details), dots of the shown colour scale – GLORIA data. Maximum potential vorticity gradient tropopause was used here.

The tracer-tracer correlation of ozone and nitric acid for the 3-D tomographic retrieval is shown in Figure 10. We can distinguish between several types of air masses in this tracer-tracer space. There is a $HNO_3$ poor tropospheric branch ($HNO_3$ < 0.2 ppbv), which consists solely of air parcels located below the dynamical tropopause (around 330 K here). A "deep stratospheric" branch conforms to the relationship of (1) within the prescribed ±15% tolerance and consists mainly of air parcels above 360 K potential temperature. Air along the flight path falls into this category based on both GLORIA and *in situ* measurement data. Most of observed stratospheric air parcels have slightly enhanced $HNO_3$ values (between +15% and +50% $HNO_3$ compared to (1)). Finally, there is the air mass with strongly enhanced $HNO_3$ values around the tropopause, clearly seen in Figure 7. It has > 1.25 ppbv $HNO_3$ and potential temperature of 330-350 K. Evidence of mixing between this air mass and its surroundings can be seen as linear structures in Figure 10.

It is clear from the argument above that the strong $HNO_3$ enhancement seen in Figure 7b-c at the altitude of around 11.5 km could not be a consequence of STE alone. The most important tropospheric source of $HNO_3$ is conversion from $NO_x$, which, in turn, is produced mostly by fossil fuel burning (24 $TgNyr^{-1}$), biomass burning (8 $TgNyr^{-1}$), soil emissions (12 $TgNyr^{-1}$) and lightning ((9 $TgNyr^{-1}$)) (Price et al., 1997; Nesbitt et al., 2000). All of these sources, except for lightning, release $NO_x$ into the boundary layer, where its lifetime often below 1 day (Tie et al., 2001) and any $HNO_3$ produced is subject to washout.



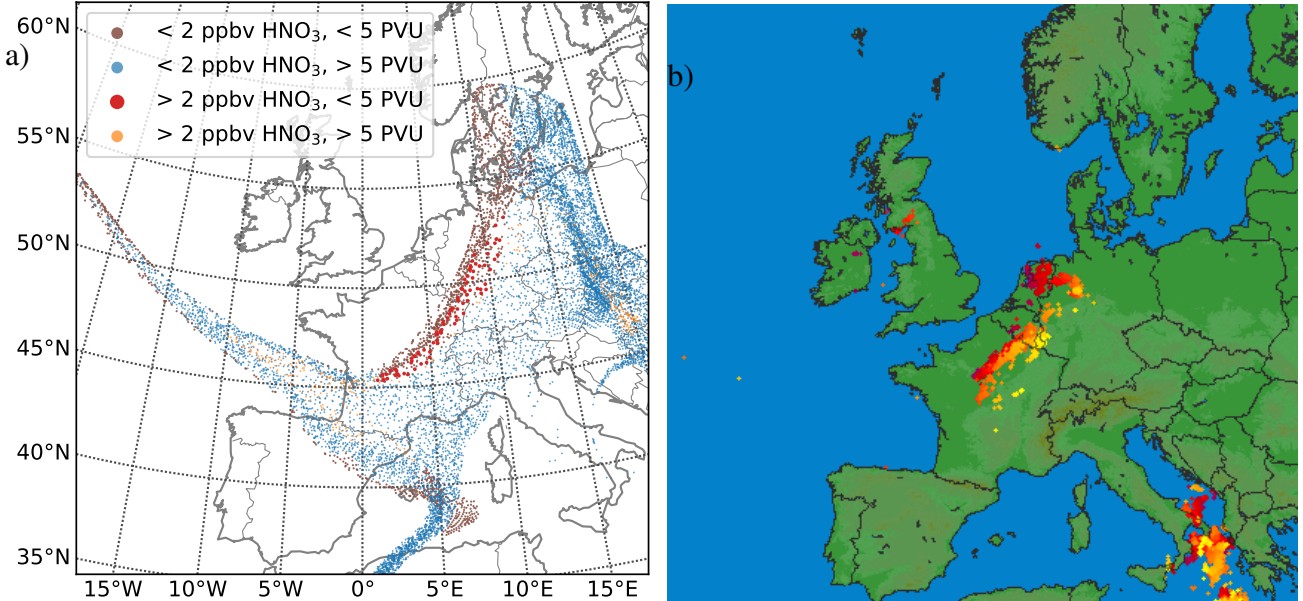

**Figure 11.** Panel a) shows the positions of air parcels observed as part of the 7 October 3-D retrieval on 29 September 18:00 UTC (8 days before measurement). Air parcel colouring: brown – less than 2 ppbv $HNO_3$, less than 5 PVU; light blue – less than 2 ppbv $HNO_3$, more than 5 PVU; red (larger markers) – more than 2 ppbv $HNO_3$, less than 5 PVU; orange – more than 2 ppbv $HNO_3$, more than 5 PVU. $HNO_3$ VMRs considered are from the 7 October 3-D retrieval. Panel b) shows lightning strikes between 29 September 18:00 UTC and 30 September 00:00 UTC in the area, based on very low frequency (VLF) radio observations. Lightning strikes are indicated by purple-to-yellow colour scale, depending on their time of occurrence within the given period.

Therefore, despite lower total emissions, lightning is the most important source of $NO_x$ in the upper troposphere (Zhang et al., 2000, 2003).

Backward trajectories were used to investigate whether the air masses with enhanced $HNO_3$ values could have been influenced by lightning. The air parcels observed on 7 October only came into proximity of significant lightning activity around 8 days before the measurement. That was also the time when the structures around the enhanced $HNO_3$ air masses were

formed, i.e., it is the earliest time when the measured air parcels constituted a relatively compact group (Figure 8). Figure 11 shows the positions of the air parcels with more than 2 ppbv $HNO_3$ and other air parcels observed by GLORIA on 29 October and the map of lightning activity that day. We see that some of the observed air parcels came into close proximity to the lightning strikes over France, Germany and the Low Countries. They are both located below 5 PVU (close to tropopause) and close by horizontally: the separation of the order of 100 km is small considering that these air masses were traced back almost

all the way around the pole over eight days. A large proportion of these air parcels are $HNO_3$ rich, and, conversely, significant proportion of $HNO_3$ rich air is located close to the lightning activity. Also, the time of 8 days between $NO_x$ emission and $HNO_3$ measurement would have been sufficient for $NO_x$ to $HNO_3$ conversion based on, e.g., Jaeglé et al. (1998). It is therefore possible, that the highest $HNO_3$ values observed on 7 October were due to lightning. The lightning positions were obtained from very low frequency (VLF) radio observations (lightningmaps.org, Narita et al. (2018)).



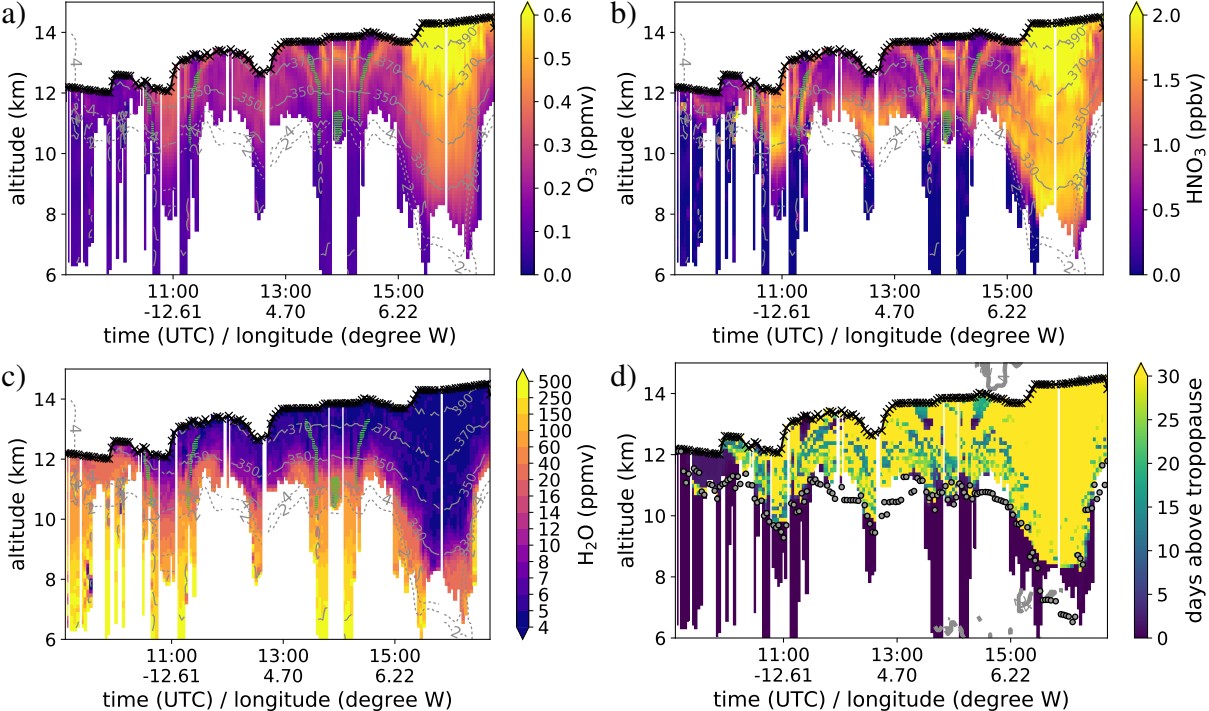

**Figure 12.** 2-D data from 9 October flight. Ozone (a), nitric acid (b), water vapour (c) retrievals with age of air plot (d) for the same air parcels. Isentropes are shown as thin solid lines, 2 and 4 PVU contours – thick dashed lines. Green hatches represent air parcels observed by GLORIA on 7 October.

### 3.4 The thin filament of the 9 October flight

The breaking Rossby wave event seen on 7 October was observed again during a flight on 9 October. No measurements suitable for tomography were taken during this flight and the 2-D data for $O_3$, $HNO_3$ and $H_2O$ are shown in Figure 12. ECMWF forecasts before the flight led us to expect that the breaking wave would be squeezed into a thin filament, shown as 2 PVU contour in Figure 1b extending from 70°N, 0°E to 51°N, 12°E (dashed magenta line). The filament was crossed twice in both directions and can indeed be seen as old air structures around 11:00 and 14:00 UTC. The old air (ozone and nitric acid rich) parts of the filament are very thin (some only around 35 km thick). The air mass that filled the hexagon (400 km in diameter) on 7 October was also stretched into a filament of 30 km thickness in its narrowest parts. Using backward trajectories the air parcels that were measured both as part of the tomographic 3-D retrieval on 7 October and 2-D retrieval on 9 October were identified. They are highlighted by horizontal green hatches in Figure 12.

As before, a plot showing the age of air in the stratosphere is provided in Figure 12d. Note the qualitatively different nature of younger air structures compared to previous plots of this sort and lack of filamentation in this region, generally less effected by planetary waves. The only sizeable young air parcels are found around the filament originating from a breaking Rossby wave. A $O_3$ - $HNO_3$ tracer - tracer plot for the air parcels observed in both flights is shown in Figure 13. The general positive





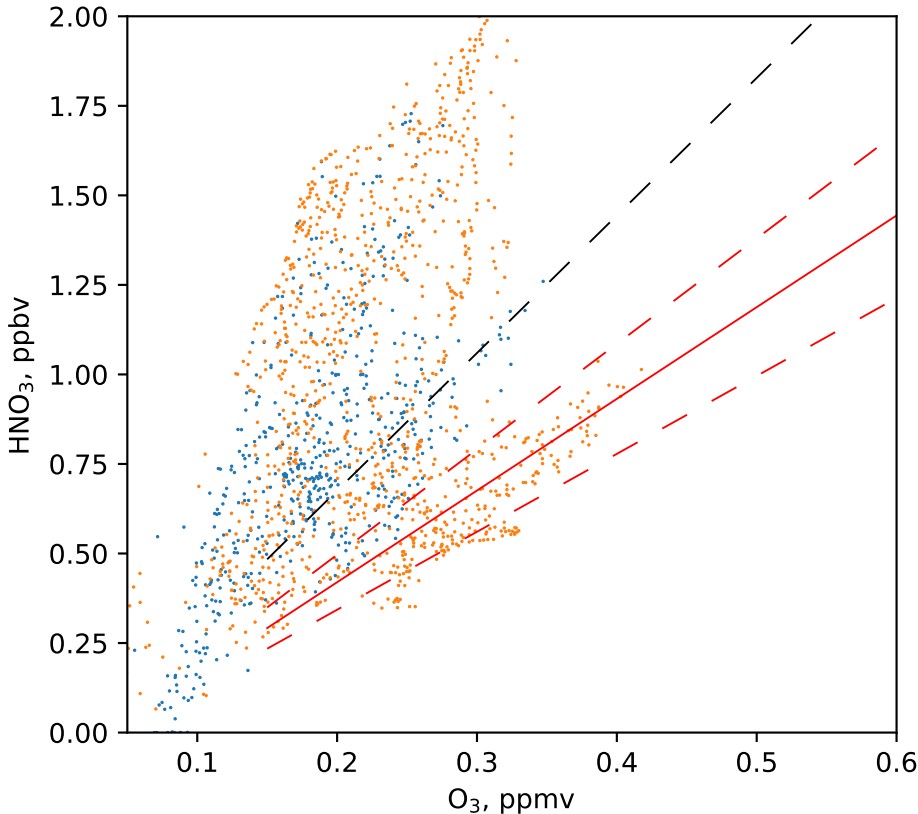

**Figure 13.** Ozone – nitric acid tracer-tracer correlation for air parcels observed in both flights. Tracer concentrations measured on 7 October (3-D tomographic retrieval) shown in orange, those measured on 9 October – in blue. Solid red line represents linear relationship of ozone and nitric acid typical for the stratosphere. Dashed red lines indicate the $\pm15\%$ $HNO_3$ compared to that relationship, black dashed line indicates +50% $HNO_3$.

correlation between $O_3$ and $HNO_3$ and the enhanced $HNO_3$ values remain in the retrieval of 9 October. Slightly more parcels

show tropospheric ozone and nitric acid values, these could be strongly diluted by tropospheric air or a result of collocation

inaccuracies. The maximum concentrations of both tracers are now slightly lower, and there are more air samples of middling

tracer values (e.g. around 0.17 ppmv of ozone and 0.65 ppbv of nitric acid), while the distinct outlying branches, such as

the highest $HNO_3$ enhancement and the purely stratospheric branch (around solid red line), are noticeably subdued. The latter

branch also appears to be shifted towards the middling values of both tracers, and is closer to the black line in Figure 13. This is

evidence of mixing, as tracer concentrations in all samples approach their average values. It is remarkable, however, that such a

thin filament preserved the general tracer-tracer structure of an air mass of completely different shape, and that relatively little

tropospheric influence is seen.

To further investigate the evolution of the air parcels observed in both flights, we determined the $HNO_3$ values in the hexagon

based on the 9 October retrieval performed 2 days later. Air parcels from the 9 October 2-D data product were traced back





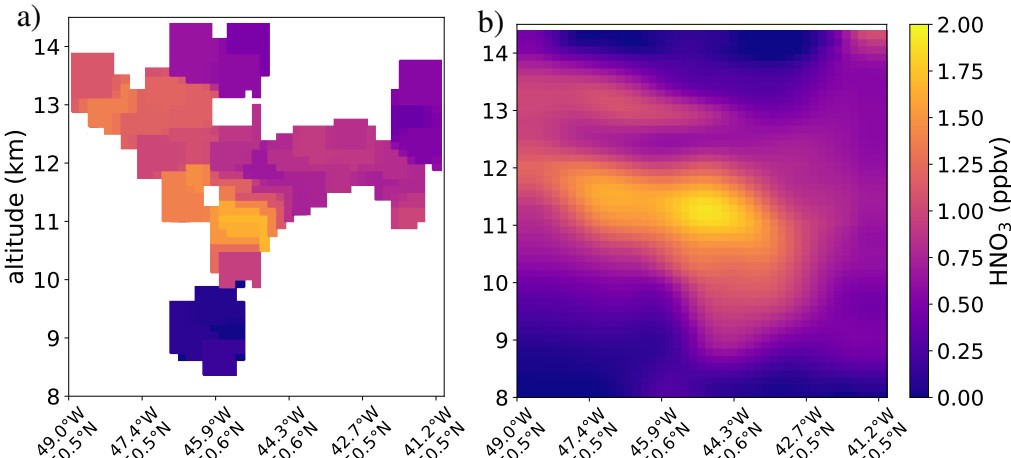

**Figure 14.** Left panel - HNO$_3$ volume mixing ratios on a vertical cut inside the hexagonal flight pattern based on back traced 9 October measurements, right panel – HNO$_3$ volume mixing ratios from the 7 October tomographic 3-D retrieval on the save vertical cut for comparison.

to the hexagon. Each grid point inside the hexagon was assigned the HNO$_3$ volume mixing ratio of the closest back traced parcel. If no parcel was traced back to a particular point within 100 km horizontally or 0.4 km vertically, the HNO$_3$ volume mixing ratio was deemed undetermined. One vertical cut through the HNO$_3$ reconstruction obtained in this manner is shown in Figure 14, with a cut through the actual 3-D retrieval from 7 October shown for comparison. We can see that the tracer structure obtained by back tracing reflects many of the features in the tomographic retrieval, just with a lot lower resolution,

since the 3-D retrieval grid has far more points than the number of air parcels that could be traced back to the 3-D retrieval volume from the observations two days later. The contrast (i.e. the difference between the highest and lowest HNO$_3$ VMRs) is lower for the back-traced data, suggesting that some mixing took place between the flights. This comparison demonstrates, however, that the thin filament, although clearly irreversibly separated from air masses around the polar jet, is still not fully mixed and "remembers" its past.

**4   Conclusions**

The UTLS in the Rossby wave surf zone is a highly interesting region of the atmosphere characterised by sharp gradients and fine structure of the tracer distributions that are very challenging to observe directly. Mixing in the Rossby wave surf zone has a major role on stratosphere-troposphere exchange (STE). Previous studies have shown that cross-tropopause transport into the extratropical lower stratosphere occurs mainly above the Atlantic and Pacific oceans and is most likely driven by Rossby wave

breaking events (e.g. Kunz et al. 2015, Vogel et al. 2016). A Rossby wave breaking event was observed with the GLORIA instrument during two research flights two days apart. Both standard 2-D data sets and 3-D tomographic data of trace gas volume mixing ratios were acquired. Signs of enhanced mixing between stratospheric and tropospheric air parcels were seen near the polar jet with some transport of water vapour into the stratosphere. The observations performed in the first flight





showed complex vertical structure in stratospheric tracers (ozone and nitric acid) with heavy filamentation, related to several
Rossby wave breaking events 3 to 8 days before observation. With the help of CLaMS backward trajectory calculations, much
of the observed tracer structure can be understood as stirring and mixing of air masses of tropospheric and stratospheric origins
(age-of-air concept). There are still unexpected aspects: for instance, the filaments were not only slanted in space, but also
stretched across different potential temperature levels. Therefore, even though these filaments were still sharply defined in
the tracer structure and not strongly mixed, they could subsequently be erased by isentropic mixing. The backward trajectory
calculations revealed large amounts of air with histories indicative of horizontal transport from the lower latitude troposphere
and matching tracer signatures were found in the observations. Air masses from the extratropical troposphere were located
mostly at the lowest altitudes of the observed part of the stratosphere. However, we found strongly uplifted (by more than
10 K) filaments, closely related to Rossby wave breaking events, and their histories were consistent with the measured tracer
content. Just above the tropopause, nitric acid was enhanced by far more than could be explained by descent of air from the
deep stratosphere. It is demonstrated in our study that $NO_x$ production by lightning activity was a likely cause of this nitric acid
enhancement. In the late stage of the RW breaking event, a filament containing air masses rich in stratospheric tracers became
very long, thin (horizontal width was down to 30 km at some altitudes) and was advected away from the polar jet. Some of
the air masses of the filament were observed during both flights (7 and 9 October) and showed a reduced contrast in tracer
VMRs in the second flight indicating some mixing of the filament with its immediate surroundings, but the tracer structures
seen during the first flight were still traceable in the data from the second. This direct observation of mixing provides valuable
data set for validation of mixing implementations in models.

A direct comparison of the full 3-D structure to model data in such detail would not have been possible with any other
observational technique. Airborne observations of a significant number of air parcels twice with two days in between the
measurements was also possible only due to the altitude range of GLORIA data and the unique 3-D capability of the instrument.

*Data availability.* All measurement data is available from the HALO database. GLORIA 3-D tomography can be found at https://halo-db.
pa.op.dlr.de/dataset/7471. GLORIA 2-D (https://halo-db.pa.op.dlr.de/dataset/5503, https://halo-db.pa.op.dlr.de/dataset/5504), FISH (https:
//halo-db.pa.op.dlr.de/dataset/5488, https://halo-db.pa.op.dlr.de/dataset/5789), AENEAS (https://halo-db.pa.op.dlr.de/dataset/5549, https://
halo-db.pa.op.dlr.de/dataset/5550) and FAIRO (https://halo-db.pa.op.dlr.de/dataset/5685, https://halo-db.pa.op.dlr.de/dataset/5687) data for
the 7 October and 9 October flights, respectively, is available using the given links.

## Appendix A: GLORIA retrieval technique

The GLORIA tomographic retrieval works by combining the JURASSIC2 fast radiative transfer model (Hoffmann et al., 2008;
Ungermann et al., 2010) with an iterative inverse modelling scheme. The main idea of this scheme can be presented as follows.
Let vector $\boldsymbol{y}$ represent a set of GLORIA radiance measurements, and let vector $\boldsymbol{x}$ represent a discrete representation of the
state of the atmosphere to be retrieved. In practice, $\boldsymbol{x}$ would contain assumed temperature and trace gas concentration values
for each point in the retrieval grid. Using the forward model, one can calculate a vector $\boldsymbol{F}(\boldsymbol{x})$ - a simulation of the radiances





GLORIA would measure, if the atmosphere were indeed in the state $\boldsymbol{x}$. Then the retrieval works by iteratively minimising the following quantity, which we call the cost function

$$J(\boldsymbol{x}) = (\boldsymbol{F}(\boldsymbol{x}) - \boldsymbol{y})^T \mathbf{S}_\epsilon^{-1} (\boldsymbol{F}(\boldsymbol{x}) - \boldsymbol{y}) + (\boldsymbol{x} - \boldsymbol{x}_a)^T \mathbf{S}_a^{-1} (\boldsymbol{x} - \boldsymbol{x}_a) \tag{A1}$$

The first term in the sum measures how strongly the simulated measurements $\boldsymbol{F}(\boldsymbol{x})$, i.e. what GLORIA would measure if atmosphere state were $\boldsymbol{x}$, differs from the actual measurement $\boldsymbol{y}$. The covariance matrix $\mathbf{S}_\epsilon^{-1}$ contains information about the instrument accuracy. Large value of this term would mean that state $\boldsymbol{x}$ is unlikely, as it does not agree with the measurements.

The second term in the sum measures how well $\boldsymbol{x}$ fits our general knowledge of the atmosphere. Here $\boldsymbol{x}_a$ is the *a priori* atmospheric state. A climatology, or low resolution model data can be used for this purpose. An atmospheric state that deviates very strongly from *a priori* is considered unlikely. The covariance matrix $\mathbf{S}_a^{-1}$ is typically non-diagonal, its purpose is not only

to rule out atmospheric states that differ from *a priori*, but also to suppress states that are discontinuous, have sharp jumps in atmospheric quantities or are otherwise unphysical. We construct this matrix to represent the exponential covariance. For two points in the atmosphere with the Cartesian coordinates $(x, y, z)$ and $(x', y', z')$, respectively ($z$ coordinate vertical), the exponential covariance between the values of an atmospheric quantity at these points is

$$C\left((x, y, z), (x', y', z')\right) = \sigma^2 \exp\left[ -\sqrt{\frac{(x - x')^2}{L_h^2} + \frac{(y - y')^2}{L_h^2} + \frac{(z - z')^2}{L_v^2}} \right] \tag{A2}$$

Hence $\sigma$ describes the magnitude the expected departure from the *a priori* for an atmospheric quantity, and $L_h$, $L_v$ determine the typical length scales of structures in horizontal and vertical directions, respectively. The relevant parameters for the retrieval in this paper are given in Table A1.

**Table A1.** Regularisation parameters for 3-D retrieval

| Parameter | Value, ppbv | Parameter | Value, km |
|---|---|---|---|
| $\sigma_{O_3}$ | 50 | $\rho_h$ | 250 |
| $\sigma_{HNO_3}$ | 0.12 | $\rho_v$ | 1 |

The standard deviations $\sigma_{O_3}$ and $\sigma_{HNO_3}$ were obtained by computing the standard deviation of in situ data for each tracer (shown in Appendix C for result validation). The ratio of horizontal and vertical correlation lengths $\rho_h/\rho_v$ was set to 250, as

this was estimated to be a typical aspect ratio for lower stratosphere (Haynes and Anglade, 1997), and also seems to match the aspect ratios of structures that form in CLaMS backward trajectory runs (Section 3.2). The final values of $\rho_h$ and $\rho_v$ were then selected to optimise the resolution of the retrieval.





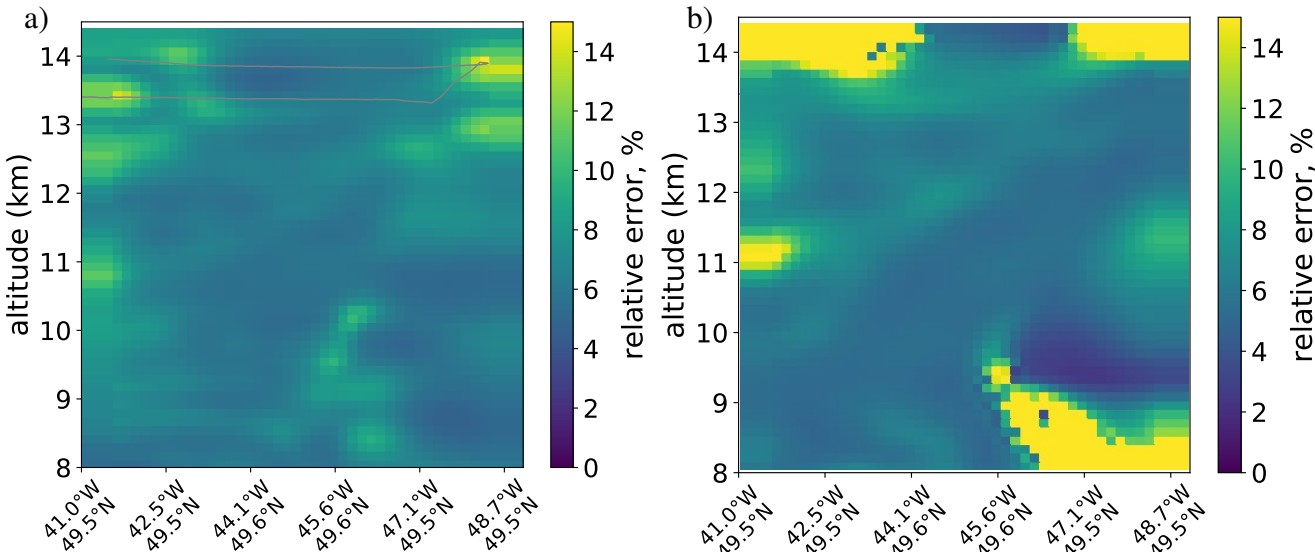

**Figure B1.** 3-D tomography relative error, Monte Carlo estimate. Panel a) – ozone, panel b) – nitric acid.

## Appendix B:  Tomographic retrieval error estimation

The data quality and expected errors of tomographic retrieval differ depending on the position of the point where these errors

are calculated with respect to the flight path (hexagonal part of pattern). The relative errors for $HNO_3$ and $O_3$ were estimated
for each point on the retrieval grid grid using a Monte Carlo based technique (Ungermann, 2013; Krasauskas et al., 2019). The
particular error estimate calculated here includes measurement error and error due to inaccuracies of the retrieval method, but
does not include errors due to imprecise knowledge of atmospheric parameters (e.g. pressure, $CO_2$ concentration, etc.), that
are required for retrieval and taken from external sources, mostly models.

Results of the calculation are presented as a vertical cut through the retrieved 3-D volume (Figure B1). The same cut as in
Figure 7, left, was used. One can confirm that, as expected, the data quality is best under the centre of the hexagonal flight
pattern and quickly gets worse just outside the hexagon. Ozone retrieval around the flight path itself also seems problematic
and should be interpreted with care. The average error values for measurements at 11 to 12 km altitude with horizontal distance
of up to 200 km from the centre of the hexagonal flight pattern are given in Table B1. The table also contains estimates for

linear retrievals at the same altitude range. They were estimated from the instrument noise and retrieval method uncertainties
in linear manner described in Ungermann et al. (2012).

**Table B1.** Average measurement error for 11 to 12 km altitude

| Retrieval | $O_3$ ppmv | $HNO_3$ ppbv |
|---|---|---|
| 7 October, 3-D | 0.014 | 0.06 |
| 7 October, 2-D | 0.006 | 0.03 |
| 9 October, 2-D | 0.008 | 0.04 |





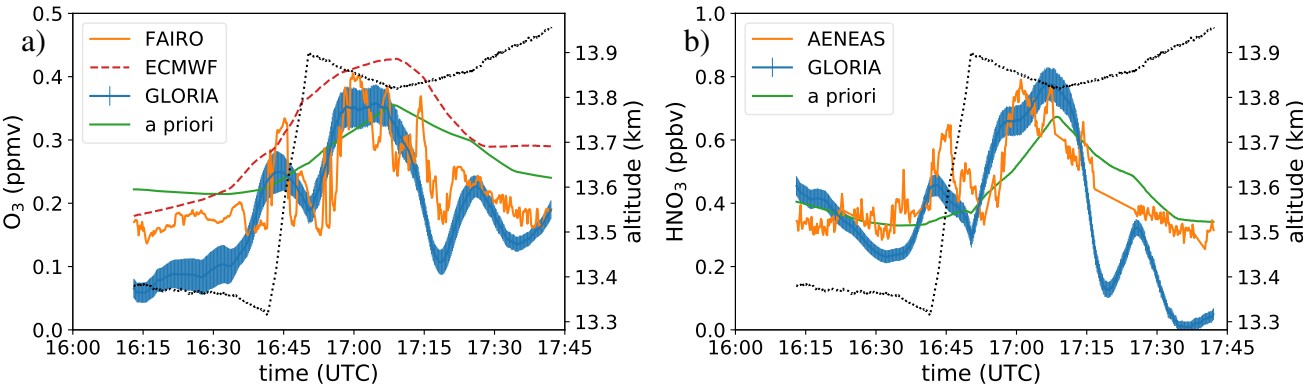

**Figure C1.** Comparison to in situ instrument data along the flight path. *A priori* data is from the WACCM model. Refer to main text regarding details on *in situ* instruments and data.

## Appendix C: Tomographic retrieval validation using *in situ* instrument data

To validate the 3-D data products, we have performed a comparison with *in situ* measurements. During WISE ozone volume mixing ratio were measured by the Fast Airborne Ozone instrument (FAIRO; Zahn et al., 2012). The Atmospheric Nitrogen
Oxides Measuring System (AENEAS; Ziereis et al., 2004) was also on board and provided $NO_y$ and NO data products. In UTLS one can generally identify the total reactive nitrogen as the sum of volume mixing ratios of the following gases (ignoring the ones with typically much lower VMRs)

$$NO_y \approx NO + NO_2 + HNO_3 + PAN + ClONO_2 \tag{C1}$$

The volume mixing rations of $NO_2$, PAN and $ClONO_2$ are all typically at least one order of magnitude lower than that of
$HNO_3$ considering the latitude and season of observation. PAN, and $ClONO_2$ can be obtained be GLORIA data, and $NO_2$ is a relatively well mixed tracer which we estimated based on WACCM model data. $HNO_3$ can hence be derived from AENEAS data for GLORIA validation purposes.

   The comparison of GLORIA data and the *in situ* instruments is presented in Figure C1. There are two important considerations that have a significant effect on how good of a match to *in situ* can be expected. Firstly, GLORIA measures trace gas
concentrations along its line of sight, with a vertical resolution of about 200 m, and horizontal resolution of 20 km. Since the whole line of sight (for usable measurements) is actually below the flight altitude, the highest altitude data available cannot be regarded as measured from the aircraft location, but rather from the location centred around $\approx 100$ m below the aircraft and $\approx 10$ km to the side of it. The result is imperfect spatial coincidence between GLORIA and *in situ*, which can introduce biases into the comparison in the areas of high tracer gradients. Secondly, the GLORIA data product is valid for 16:50 UTC, while the
*in situ* data is valid for the measurement time (see advection compensation description in Section 2.1). The data in Figure C1 is corrected for this discrepancy in the sense that GLORIA data is provided for the actual air masses that the aircraft was flying through and not for the locations with respect to ground that were sampled. By 16:50 UTC, however, many of these air masses,



especially the ones from the beginning of the flight, were advected outside of the best resolved area of the hexagonal flight path (i.e. there were few overlapping measurements of these air parcels). This has a negative effect on data quality there. The errors

for air further below flight altitude and closer to the centre of the hexagon should therefore be more accurate.

*Author contributions.* PP, MR, FP and BV contributed to flight planning. FFV coordinated GLORIA operations. JU performed the level 1 data processing and prepared the 2-D GLORIA data products. LK prepared the 3-D GLORIA data products, performed backward trajectory analysis, tracer correlation analysis and wrote most of the manuscript. AZ provided the ozone data. CR provided FISH data. HZ provided the NOy and NO data used as a proxy for $HNO_3$. PP, JU, FP, PK, MR, CR and BV contributed many ideas for towards the different aspects

of data analysis and interpretation. All authors commented on the paper and helped to improve it.

*Competing interests.* The authors declare that they have no conflict of interest.

*Financial support.* Lukas Krasauskas was partly funded by the German Science Foundation (Deutsche Forschungsgemeinschaft, DFG) under the DFG project AMOS (HALO-SPP 1294/VO 1276/5-1). The WISE campaign was supported by the German Research Foundation (Deutsche Forschungsgemeinschaft, DFG Priority Program SPP 1294).

*Acknowledgements.* The authors gratefully acknowledge the computing time granted through JARA on the supercomputer JURECA at Forschungszentrum Jülich. The European Centre for Medium-Range Weather Forecasts (ECMWF) is acknowledged for meteorological data support. Blitzortung.org and contributors and LightningMaps.org are credited for the lightning data. The authors especially thank the GLORIA team, including the institutes ZEA-1, ZEA-2 at Forschungszentrum Jülich and the institute for Data Processing and Electronics at the Karlsruhe Institute of Technology, for their great work during the WISE campaign on which all the data in this paper are based. We

would also like to thank the WISE flight planning team as well as the pilots and the ground support team at the Flight Experiments facility of the Deutsches Zentrum für Luft- und Raumfahrt (DLR-FX).





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
