# Peer review of "3-D tomographic observations of Rossby wave breaking over the Northern Atlantic during the WISE aircraft campaign in 2017"

_Atmospheric Chemistry and Physics, 2020_

## Short Comment (SC1) · 19 Jan 2021

The paper presents measurements of trace gases taken during a Rossby wave breaking event using the airborne limb sounding instrument GLORIA. This paper was discussed at the last meeting of the SPARC Stratospheric Dynamics journal club*. We thought that the methodology employed yielded interesting results and have summarised some comments that came up during our discussion. This is not a full review.

Overall comments: The figures have a high density of information and it was often difficult to find the specific features being referred to. This could be improved by highlighting the specific features, referring to them in the caption and adding a figure key in

addition to the long descriptions in the caption. For examples in Figure 1, a figure key for the lines, dots and other contours would be useful.

Methods: Describe the FISH and FAIRO instruments. Where is PV data from? We assumed it was from the ECMWF operational analysis.

Figure 1: What do the yellow dots with black outlines along the limb view represent? Colours for the chemistry and dynamics mode could be more distinct. The yellow looks pale green.

Line 168: "O3 and HNO3 are more abundant in the stratosphere, where they are generated by photolysis" Is HNO3 generated in this way? I thought it was photolysed into NO2

Figure 4: The colour scale makes the air masses hard to interpret. Perhaps a different one would work better. We found the features being referred to such as the double mixing lines unclear. Annotations would be helpful too. Would it be possible to give a rough number of observations plotted in the caption for all such figures?

Figure 6 was hard to visualise. Would a rotated perspective work better? It would be good to also have latitudes and longitudes along the horizontal axes and to indicate the cross-sections from Figure 7 here.

Figure 8, label 165 W. We were also confused about how the calculation was done. Were the regions in Figure 7 (c) and (d) selected prior to doing the back trajectories or were they found after Figure 8 (a) showed that there were two groups of particles – red and black?

Line 311 - Where it reads 29 October, it should read 29 September.

Figure 9 (c) was not referred to in the text. Please describe what this figure shows.

——- Alison Ming, Corwin Wright, Elio Campitelli, Inna Polichtchouk, Timothy Banyard, Annelize van Niekerk + 1 other

ACPD

*We are a group of about 25 members across about 12 different institutions. A subset of us meets every 2 weeks to discuss a paper.

---

## Referee Comment (RC1) · Anonymous Referee #1 · 29 Jan 2021

The paper presents novel observations of the three dimensional tracer structures during subsequent measurements of a Rossby Wave breaking event. Ozone and HNO3 observations are obtained from the GLORIA -instrument onboard the HALO aircraft during the WISE mission in September / October 2017 over the Atlantic.

The authors first present an analysis of the two dimensional tracer structure during the wave breaking parallel to the flight track. Data cover curtains of 2-6 km below the aircraft flight altitude. The authors focus on the distribution of H2O , ozone and HNO3 to infer signatures of cross tropopause mixing. The tracer observations indicate a very rich tracer structure in the tropopause region indicating a highly complex dynamical

history of transport and mixing. Using the chemical Lagrangian Model of the Strato-sphere (CLaMS) the authors analyze transport time since tropopause crossing which they can link to the observed tracer structures. The highlight and core of the paper is the three dimensional view on the tracer structure, which they study to derive trans-port and mixing histories for the different tracer filament. They analyze the apparent filsmentes by a comprehensive use of CLaMS information and show the complex di-abatic history of the encountered filaments and air parcels. They particularly show a complex interplay of diabatic processes which is remarkably well represented in the tracer structure indicating, that the mixing process is rather inefficient preserving the chemical separation of air masses surprisingly long. On a consecutive flight two days later they could trace back the stirred filament via CLaMS trajectories to the 3D tracer structure two days before. Notably they could trace back the chemical anomalies to the tomographic volume, which is another highlight result. It shows, that the mixing time scale is slow allowing still to have chemically distinct regimes after several days of stratospheric residence times.

All in all the results presented here are clearly novel and clearly merit publication in ACP. The manuscript contain the analysis of an unprecedented 3-D view on the effect of RWB on the chemical structure of the lower stratosphere and the diabatic changes which are associated with this. This last process-based aspect should be a bit more elaborated. To make it a highlight the authors should exploit and extend their analysis of Figs.6-9, since this is the first comprehensive 3-D view on such an event. With CLaMS they could easily get closer to the cross tropopause exchange process than just stating RWB), by e.g. analyzing mixing strengths, driving factors of diabatic changes along the trajectory etc. They could get much more out of the analysis especially Fig.6-9). There are also a lot of slang-like expressions and a terminology, which is qualitative or non-scientific which should be changed to a more concise scientific wording. After these revisions have been applied I highly recommend the manuscript for publication.

General points: Terminology: - replace age or age of air by 'stratospheric residence

time' or 'statospheric transit time' - use the terminology established by e.g. Stohl et al., 2003: - troposphere-to-stratosphere-transport (TST) - stratosphere-to-troposphere-transport (STT) - stratosphere-troposphere-exchange' (STE including both TST, STT)

The analysis of the 3D history in Figs.6-9 could be sharpened by analysing for the (diabatic) processes which lead to diabatic changes and TST (and distinguish from quasi-isentropic exchange). It allows determining the complex interplay between different processes and should be really stressed a bit more as pointed out above. - The analysis of diabatic changes and tropopause crossings are really great, is it possible to deduce where and by which process diabatic ascent was produced (frontal uplift, WCB,...?) in contrast to more isentropic transport (e.g. for exchange at hight Theta values)? - Fig 9c) is remarkable, but are the processes creating the distinct TST maxima the same or is the upper part from quasi-isentropic TST? Is the maximum number at lower Theta due to midlatitudinal synoptics (again more diabatic TST: WCB, frontal uplift in mid latitudes...)?

Further, as indicated below more specifically I missed isentropic PV maps to diagnose mixing. It's clear, that the native coordinate of aircraft and observation is geometric, but the analysis of dynamical features and mixing should also be done analyzing isentropic PV maps, particularly when looking at TST.

Specific: Could you add in one of the cross sections in Fig. 2 the horizontal wind speed to motivate the classification of 'near jet' and 'away from jet' relative to the cross sections?

l.104-106: This sentence is weird, rephrase.

l.125: Replace 'were executed' by e.g. 'were flown' or similar.

l.150 and whole paper: replace 'Age of air' by stratospheric residence time' or 'stratospheric transit time'

l.166: The statement about water vapor holds for the extratropics. The upper tropospheric part of the TTL can be very dry (<10 ppmv) as well, which is important for exchange at high potential temperatures.

l.168: Ozone and HNO3 are not produced by photolysis, better use 'photochemistry'.

l.177. The mixing time scale is an completely open issue and I wonder, if this manuscript using the 3D information from GLORIA and the mixing parametreisation of ClaMS can further quantify these mixing time scales? This could be a really novel aspect.

l.181: Though the authors clearly indicate their use of the term 'age' I highly recommend to replace it by 'stratospheric residence time' or 'stratospheric transit time' since the term 'age' is used for the mean stratospheric age of air (i.e. the mean of the transit time distribution of individual stratospheric air parcels).

l.180-185: The Figure 2d is great, but also puzzling, since it implies tropospheric impact all over the curtain with residence times from 0 to 30 days. Could the authors provide a complementary figure with the fraction or amount of trajectories staying in the stratosphere? This would further support the potential impact of TS (troposphere-to-stratosphere-transport)

l.191 'a'ffected

l.208: The use of water vapor to identify stratospheric air masses is ambiguous since in the tropical and subtropical upper troposphere low water vapor below 10 ppmv at low ozone levels also show up leading to mixing between stratospheric and TTL air (e.g. greenish in the lower left quadrant of Fig.4a). The opposite, however, holds (and is important for the paper): enhanced water vapor clearly indicates tropospheric contributions from mid and high latitudes (e.g. 4b) 5) and the upper right quadrant clearly shows mixing. Is it possible to use this also to support the trajectory analysis in Fig. 9a,c)?

Caption Figs. 4/5: Distance to the PV-gradient-derived, the dynamical tropopause or

the thermal tropopause?

Line 216- 218.: How do you infer an 'influx' of stratospheric air into the UT? This would imply stratospheric water of >20ppmv, which is unrealistic. Do you mean influx of stratospheric air (as in l218-220)? The two branches seen in GLORIA in Fig.4a seem to indicate mixing into the stratosphere (i.e. to ozone values above 100 ppmv) from different source regions: To check this a second plot using simply potential temperature as color would be helpful. In case of different isentropic source regions, this should show up.

Is it possible to indicate these airmasses (branches in Fig 4a) in one of the curtains in Fig.2? A discrete color bar in Figs. 4/5 would help.

How does the stratospheric residence time (from Fig.2d) look as color code in the correlations (Fig.4)? Mixing of distinct air parcels may show up and would indicate eventually a mixing time scale (or provide an upper limit).

l.223: Not necessarily uplift ,could be isentropic transport as well.

l.256: What is meant with 'the retrieval sampled...'? Better rephrase

l.262: The old and young air masses indicated by dark colors in Fig.7...

l.280: Which air masses are meant with isentropically mixed (in Fig9)? The continuous color code is not easy to read. See previous comments: I think this could be elaborated a bit more, which trajectories of those in Fig.9a came from the PBL, which from the TTL (e.g. color coding max/ pressure of TST-trajectory), this should also help to distinguish rapid uplift from quasi-horizontal exchange ?

l.313: The Netherlands (instead of 'low countries')

l.311: 29 September (replace October)

l.323: '….. RWB squeezed into a thin filament sounds weird.

l.324: dashed magenta line: I can only find one in Fig. 1a)

l.326/327: Avoid 'thin' and 'thick' in this context, that's non-scientific.

l.332: Whats a sizeable air parcel? Change term.

l.336: and later 'middling' - whats middling?

L3.67: 'Age-of-air -concept': must be removed here, since this is a reserved term for stratospheric age.

Figures in general: The use of continuous color bars is not always useful, e.g. the comparison of Figs 9 and 7b) is difficult.

Fig.1: Could you present the evolution of the filament crossing the hexagon on isentropic PV maps from 7.Oct to 9. October in steps of 12 hours (eventually, not necessarily, for the appendix)? Isentropic PV maps are commonly used to track dynamics and would facilitate the discussion and cross sections of the second flight.

Figs.2/12: It would be helpful to provide vertical cross sections of PV (discrete color), Theta- and windspeed contours as one additional cross section.

Refs: Stohl, A., Wernli, H., James, P., Bourqui, M., Forster, C., Lin-iger, M. A., Seibert, P., and Sprenger, M.: A new perspectiveof stratosphere-troposphere exchange, Bull. Am. Met. Soc., 84,1565–1573, 2003.)

---

## Referee Comment (RC2) · Anonymous Referee #2 · 2 Feb 2021

Review of: 3-D tomographic observations of Rossby wave breaking over the Northern Atlantic during the WISE aircraft campaign in 2017.

By Krasauskas and colleagues

—- General comments

This is a very nice paper that does a thorough job of capitalizing on information from a state-of-the art measurement system (instrument and retrieval algorithms) to provide insights into fine scale processes at work in the extratropical upper troposphere and lower stratosphere. The work is solid and well described, the data and models support the conclusions reached, the work is a welcome addition to the body of knowledge,

and the standard of the writing and figures are both excellent.

I really have no "large scale" suggestions for improvements to make about the paper, and I'm very happy to recommend that it be ultimately accepted, pending some very minor clarifications and suggestions detailed below. I look forward to seeing this paper in press.

—- Minor comments

As I say, all these are minor suggestions for improved wording, clarity, clarification, etc.

Line 17: "witch" -> "which"

Line 22: "The composition of +the+ UTLS", also, add "and" at the end of this line.

Line 24/25: Move "(RW)" from line 25 to right after "Rossby wave" on lien 24.

Line 27: "from +the+ troposphere"

Line 85: I'm not sure "tomography images" is really the right word. Firstly, I'd suggest "tomographic" rather than "tomography". But regarding "images", those unfamiliar with retrievals might confuse them with the Level-1 radiance "images" (which really are "images", in the traditional 2-D sense of the word). "Fields" is an alternative, but I'm it's not quite right either, I recognize. Others might be "depictions", "representations", but I'm not sold on them either. Anyway, something to ponder.

Line 90: At face value, the discussion here seems to be talking about "high vertical resolution" for the radiances, but I think you mean it to apply also to the retrieved state, correct? It might be good to add a few words to clarify that "...not only for the measured radiances, but also, ultimately for the retrieved atmospheric states corresponding to those measured radiances".

Line 112: Is the temperature also "advected"? If so, how, as some kind of tracer? Is that valid meteorologically speaking?

Figure 1: I found the yellow hard to spot, it's more of a lime green on my printer.

Line 162: "In the case of cloud presence..." -> "In cases where clouds are present, atmospheric properties can only be retrieved in the regions above the cloud tops".

Line 164 (and 162 before the above edit): "data" is plural (it's the plural of "datum"). So "is" -> "are". There may be other places where this needs to be fixed, I didn't check exhaustively.

Line 167: Put an "e.g." before the Schiller citation.

Line 168: If you're including a citation for ozone (line above) why not one for HNO3 also, for symmetry.

Figure 2: It's a bit odd that panel (a) has a discrete color scale while (b)-(d) are continuous. I tend to favor the discrete ones myself, as that makes filaments more clear, but either way, it might be better to be consistent.

Line 171/172: "Retrievals of H2O, O3, and HNO3 confirm this generally expected behavior, but also..."?

Figure 4: The captions should note what the dashed black lines signify. (e.g., "the 0.1 and 10 ppmv values for O3 and H2O, respectively, as discussed in the text" or something like that).

Lines 199/200: "The air found near...3 days at least". Are we supposed to be able to see that from examination of Figure 3? If so, it wasn't clear to me, so perhaps more hand-holding is required.

Line 201: Could/should the word "subsequently" be inserted right before "transported" (final word in this line). If that's not correct, then that means I haven't understood the discussion properly, and perhaps some clearer discussion is needed.

Figure 6: I'm afraid I found this figure hard to interpret/visualize. Perhaps, rather than multiple filled contours oriented vertically, might it be better to have just a few colored

contour lines slicing horizontally. (e.g., Figure 6 doi:10.1002/2015JD023488).

Figure 7: I had to really search for the solid grey line. Could you make it thicker (and perhaps paler?)

Line 256: "This is because" is slightly weak wording. Firstly, it's not clear what the "This" refers too. Perhaps just saying "Specifically" would be better?

Lines 257, 260, 263: I don't think the use of the colons here is quite correct. In most cases, I think just starting a new sentence would be better. Also, for the double quotes here you should probably distinguish the open and close quotes.

Figure 9: Panel (a) is not discussed in the text so far as I could see. Also, panels (b) and (c) are discussed in reverse order. Consider discussing (a) and swapping the order of the panels so that they discussion and figure orders agree.

Line 273/274: "Panel c) shows the distribution of the potential temperatures at which the observed air parcels entered the stratosphere (maximum..."

Line 287/288: "Owing to the similarity of their source and sink regions, HNO3 and O3 typically display a very compact relationship within the stratosphere.

Line 304: Add "is" after "lifetime"

Line 401: "Large value+s+ of this term...". I think you should make it clear that (a) "this term" is J(x), correct, and also that (b) you mean large values after the iteration has converged, right?

That's it! Very nice job!

---

## Short Comment (SC2) · 8 Feb 2021

Lukas Krasauskas

l.krasauskas@fz-juelich.de

Thank you for the helpful comments and interest in the manuscript. We will try to address your overall comments in the next revision. The answers to the more specific questions and comments are given below. The questions themselves are quoted in italic.

*Describe the FISH and FAIRO instruments. Where is PV data from? We assumed it was from the ECMWF operational analysis.*

The PV data is indeed from the ECMWF operational analysis.

[Figure]

The measurement principle used by the Fast In-Situ Stratospheric Hygrometer (FISH) is based on photofragment fluorescence: water molecules are split into an excited OH molecule and a single H atom by Lyman-$\alpha$ radiation (121.6 nm). The excited OH molecules emit radiation in the 285–330 nm range when relaxing to the ground state. This emitted radiation is detected by a photomultiplier tube. The number of detected fluorescence photons is proportional to the water vapor mixing ratio with a calibration factor. This calibration factor is determined prior to each experiment in the laboratory, using the commercial frost-point hygrometer DP30. FISH data with 1 Hz sampling was available for the WISE campaign. The instrument is described in detail in Meyer et al. (2015), as cited in the manuscript.

The Fast Airborne Ozone instrument (FAIRO) is based on the chemiluminescence of ozone (at $\lambda$ = 450-500 nm) on the surface of an organic dye (Coumarin 47) adsorbed on silica gel powder deposited on an aluminum disk. The chemiluminescence light is detected by a small channel photomultiplier. The instrument is calibrated using UV photometry. The measurement frequency when flying aboard the HALO aircraft is 12.5 Hz, which results in a high spatial resolution of 20-25 m at cruising altitude. The instrument is described in detail in Zahn et al. (2012), as cited in the manuscript.

*Figure 1: What do the yellow dots with black outlines along the limb view represent? Colours for the chemistry and dynamics mode could be more distinct. The yellow looks pale green.*

Colours will be updated in the next version of the manuscript. The yellow dots with black outlines represent the tangent points in each profile that are closest to the altitude shown in the plot (11 km for Figure 1a, 10 km for Figure 1b). This will be included in the figure description. The remaining tangent points are shown in green.

*Line 168: "O3 and HNO3 are more abundant in the stratosphere, where they are generated by photolysis" Is HNO3 generated in this way? I thought it was photolysed into NO2.*

[Figure]

Thank you for pointing this out, the statement you mentioned is indeed imprecise. It was meant to state that "O3 and HNO3 are more abundant in the stratosphere, where their VMRs are controlled by photolysis".

*Figure 4: The colour scale makes the air masses hard to interpret. Perhaps a different one would work better. We found the features being referred to such as the double mixing lines unclear. Annotations would be helpful too. Would it be possible to give a rough number of observations plotted in the caption for all such figures?*

The figures were updated with a new colour scale which has better contrast between points (see Figures 1-3 in this document, replacing Figures 4a, 4b and 5, respectively, of the manuscript). The double mixing lines in the comparison of Figures 4a and 5, admittedly, should have been explained better. They refer to the two clusters of air parcels in the 10-30 ppmv $H_2O$ range, now highlighted by black lines in Figure 1, and seen in Figure 3 as well. All these changes will also be implemented in the new manuscript. The number of GLORIA observations for Figure 4a, 4b and 5 are 12706, 25359 and 5032, respectively, with majority of these points located in purely tropospheric and purely stratospheric regions of the tracer-tracer space.

*Figure 6 was hard to visualise. Would a rotated perspective work better? It would be good to also have latitudes and longitudes along the horizontal axes and to indicate the cross-sections from Figure 7 here.*

Figure 6 will be replaced by an improved version, given in this document as Figure 4. Black line there represents the flight path, blue line – the horizontal projection of flight path.

*Figure 8, label 165 W. We were also confused about how the calculation was done. Were the regions in Figure 7 (c) and (d) selected prior to doing the back trajectories or were they found after Figure 8 (a) showed that there were two groups of particles – red and black?*

[Figure]

The regions corresponding to the red and black groups of particles are shown in Figure 7 (e) and (f). I am assuming you meant to ask about these, since Figure 7 (c) and (d) do not highlight any regions. These regions were defined after the analysis on which Figure 8 (a) is based showed that the particles were divided into two groups until shortly before the measurements.

*Figure 9 (c) was not referred to in the text. Please describe what this figure shows.*

Figure 9 (c) was referred to in the text. Here is an excerpt of the manuscript describing the figure: "Figure 9 gives some insight into the vertical transport of observed air parcels. Panel c) shows the distribution of the observed air parcels according to the potential temperature at which they entered stratosphere (maximum potential vorticity gradient tropopause was used to determine the entry point). It shows the two distinct pathways of air into the stratosphere around the polar jet: from low potential temperature levels ($\approx$ 340 K or less) upwards across the tropopause, or isentropically from lower latitudes at high potential temperature (horizontal transport). The latter pathway plays a major role, as expected for this region (Holton et al., 1995; Pan et al., 2009)."
* * *
**Fig. 1.** Ozone – water vapour tracer-tracer correlations from western part of 7 October flight. Black dots represent in situ measurements at flight altitude, black lines - mixing lines referred to in the text.

**Fig. 2.** Ozone – water vapour tracer-tracer correlations from eastern part of 7 October flight. Black dots represent in situ measurements at flight altitude.

**Fig. 3.** Ozone – water vapour tracer-tracer correlations from 9 October flight. Black dots represent in situ measurements at flight altitude.

[Figure]

**Fig. 4.** 3-D plot of tomographic retrieval of ozone VMR. Flight track represented by thin black line, a horizontal projection of slight track – by thin blue line.

---

## Author Comment (AC1) · 30 Mar 2021

We thank the reviewer for the comments and suggestions that helped to improve this paper. We are especially grateful for the ideas on mixing analysis, that helped to extend the relevant part of the manuscript.

The reply is given below. We do not discuss small technical or typesetting remarks and typos spotted by the reviewer here, those were simply applied as recommended. The original reviewer comments are indented, exerpts from the revised version of the paper are given in italic.

General comments

> Terminology: - replace age or age of air by 'stratospheric residence time' or 'statospheric transit time' - use the terminology established by e.g. Stohl et al., 2003: - troposphere-to-stratosphere-transport (TST) - stratosphere-to-troposphere-transport (STT) - stratosphere-troposphere-exchange' (STE including both TST, STT).

The use of "age of air" was replaced throughout the paper.

> The analysis of the 3D history in Figs.6-9 could be sharpened by analysing for the (diabatic) processes which lead to diabatic changes and TST (and distinguish from quasi-isentropic exchange). It allows determining the complex interplay between different processes and should be really stressed a bit more as pointed out above. - The analysis of diabatic changes and tropopause crossings are really great, is it possible to deduce where and by which process diabatic ascent was produced (frontal uplift,WCB,...?) in contrast to more isentropic transport (e.g. for exchange at hight Thetavalues)? - Fig 9c) is remarkable, but are the processes creating the distinct TST maxima the same or is the upper part from quasi-isentropic TST? Is the maximum number at lower Theta due to midlatitudinal synoptics (again more diabatic TST: WCB, frontal uplift in mid latitudes...)?

The analysis mentioned here was extended by including air mass origin (TTL, PBL, extratropical troposphere) analysis and uplift locations into the former Figure 9 (Figure 10 in the new version), and expanding the relevant discussion in the main text. In short, the higher theta TST maximum is almost entirely due to isentropic transport from the TTL. The lower theta maximum exists because of the transport from extratropical upper troposphere, and has contributions both from the TTL and extratropics. The direct PBL contribution (i.e. without passing the TTL first) was rather small for the 3D data set,

explaining the relative lack of water vapour in the hexagonal part of the flight.

> Further, as indicated below more specifically I missed isentropic PV maps to diagnose mixing. It's clear, that the native coordinate of aircraft and observation is geometric, but the analysis of dynamical features and mixing should also be done analyzing isentropic PV maps, particularly when looking at TST.

PV maps at 340 K potential temperature levels were added as the new Appendix D, as requested here and in the more specific comments below.

Specific comments

> l.166: The statement about water vapor holds for the extratropics. The upper tropospheric part of the TTL can be very dry ($<$10 ppmv) as well, which is important forexchange at high potential temperatures.

The statement was corrected to *"Generally, water vapour has high volume mixing ratios in the extratropical troposphere [..]"*. The paragraph in question is mostly relevant to the regions where we measured (far from the tropics), the role of the TTL is discussed separately.

> l.177. The mixing time scale is an completely open issue and I wonder, if this manuscript using the 3D information from GLORIA and the mixing parametreisation of ClaMS can further quantify these mixing time scales? This could be a really novel aspect.

Mixing analysis that would use GLORIA data and the ClaMS mixing scheme is something we would like to do in the future, but we think it is out of scope of this paper. For

now, we did extend the mixing analysis using the ideas in the reviewer comment on lines 216-218 (see below) to gain some insight into mixing time scales.

l.180-185: The Figure 2d is great, but also puzzling, since it implies tropospheric impact all over the curtain with residence times from 0 to 30 days. Could the authors provide a complementary figure with the fraction or amount of trajectories staying inthe stratosphere? This would further support the potential impact of TS (troposphere-to-stratosphere-transport)

In Figure 2d the air parcels with stratosphere residence time of 30 days or more are all depicted in the same colour (the clarification was added to the plot). We thought it appropriate, because we expect any tracer structures due to STE to be erased by that time (Juckes and McIntyre, 1987, as cited in the paper). Our results seem to generally confirm this, tracer contrast is seen between air masses with smaller residence times. The figure, therefore, does not imply tropospheric impact all over the curtain with residence times from 0 to 30 days, many trajectories originate from stratosphere. It just does not seem very meaningful to distinguish between the high residence times in this context.

l.208: The use of water vapor to identify stratospheric air masses is ambiguous since in the tropical and subtropical upper troposphere low water vapor below 10 ppmv at low ozone levels also show up leading to mixing between stratospheric and TTL air(e.g. greenish in the lower left quadrant of Fig.4a). The opposite, however, holds(and is important for the paper): enhanced water vapor clearly indicates tropospheric contributions from mid and high latitudes (e.g. 4b) 5) and the upper right quadrant clearly shows mixing. Is it possible to use this also to support the trajectory analysis in Fig. 9a,c)?

The simple classification described here serves the purpose of introducing the reader

to the tracer-tracer correlations and, we believe, is appropriate for identifying STE-related mixing that occurs in the region where we measured (we did not measure in the tropics). The more detailed understanding of air mass origins and possible mixing within the stratosphere does indeed require further analysis, which was attempted with the new Figures 4c, 5 and the accompanying discussion.

The Figure 9 (old version of manuscript) is based on the 3D data set. The region inside the hexagon, sadly, did not contain the interesting water vapour structures seen elsewhere in the stratosphere with the help of 2D retrievals.

> Line 216- 218.: How do you infer an 'influx' of stratospheric air into the UT? This would imply stratospheric water of >20 ppmv, which is unrealistic. Do you mean influx of stratospheric air (as in l218-220)? The two branches seen in GLORIA in Fig.4a seem to indicate mixing into the stratosphere (i.e. to ozone values above 100 ppmv) from different source regions: To check this a second plot using simply potential temperatures color would be helpful. In case of different isentropic source regions, this should show up. Is it possible to indicate these air masses (branches in Fig 4a) in one of the curtains in Fig.2? A discrete color bar in Figs. 4/5 would help.How does the stratospheric residence time (from Fig.2d) look as color code in the correlations (Fig.4)? Mixing of distinct air parcels may show up and would indicate eventually a mixing time scale (or provide an upper limit).

We are especially grateful for this insightful comment, as it gave ideas for extending our mixing analysis. We believe that the high ozone values at low potential temperatures mentioned in Lines 216-218 were caused by a partially mixed air mass descending from the stratosphere. See Figures 4 and 5 in the new version of the manuscript, as well as the additions to Section 3.1 for more detailed discussion based on backward trajectories and tracer correlations. The new analysis also gives some insight into mixing time scales.

l.223: Not necessarily uplift ,could be isentropic transport as well.

The sentence was reformulated, referring to the new mixing analysis.

l.280: Which air masses are meant with isentropically mixed (in Fig9)? The continuous color code is not easy to read. See previous comments: I think this could be elaborated a bit more, which trajectories of those in Fig.9a came from the PBL, which from the TTL(e.g. color coding max/ pressure of TST-trajectory), this should also help to distinguish rapid uplift from quasi-horizontal exchange.

This question was addressed by extending Figure 9 (now Figure 10) to indicate the regions of air mass origin and adding some backward trajectory examples as the new Figure 9, which would hopefully make the role of RWB clearer.

l.324: dashed magenta line: I can only find one in Fig. 1a)

This was a typo, the dashed line is black.

---

## Author Comment (AC2) · 30 Mar 2021

We thank the reviewer for the comments and suggestions that helped to improve this paper.

The reply is given below. We do not discuss small technical or typesetting remarks and typos spotted by the reviewer here, those were simply applied as recommended. The original reviewer comments are indented, excerpts from the revised version of the paper are given in italic.
Specific comments

> Line 90: At face value, the discussion here seems to be talking about "high vertical resolution" for the radiances, but I think you mean it to apply also to the retrieved state,correct? It might be good to add a few words to clarify that "...not only for the measured radiances, but also, ultimately for the retrieved atmospheric states corresponding to those measured radiances".

This statement could indeed be misleading. It was clarified: *"[...] high vertical resolution of up to 200 m of the retrieved atmospheric quantities."*

> Line 112: Is the temperature also "advected"? If so, how, as some kind of tracer? Is that valid meteorologically speaking?

This is indeed an important issue. To retrieve trace gas concentrations, temperature needs to be retrieved as well, and, unlike trace gases, one cannot assume it is simply advected. The solution for this issue is a topic of ongoing work, it is important for observations of phenomena with strong, small scale temperature disturbances, such as gravity waves. Fortunately, no such phenomena were present in this case, the retrieved temperature field was relatively smooth and compared well with in situ observations along the flight path. This was deemed sufficient for the trace gas products presented here. The uncertainties due to temperature retrieval are included in the error analysis in Appendix B.

> Line 168: If you're including a citation for ozone (line above) why not one for HNO3 also, for symmetry.

The corresponding topic for nitric acid is covered in Popp et. al. (2009), which is cited at the end of the next sentence.

Figure 2: It's a bit odd that panel (a) has a discrete color scale while (b)-(d) are continuous. I tend to favor the discrete ones myself, as that makes filaments more clear, but either way, it might be better to be consistent.

Panel a, unlike the others, uses a logarithmic scale. We feel that it is helpful to add as many labels as possible to the colour bar in this case. Round-number values, however, end up unequally spaced next to continuous logarithmic scales, which would make the bar very hard to read.

Lines 199/200: "The air found near...3 days at least". Are we supposed to be able to see that from examination of Figure 3? If so, it wasn't clear to me, so perhaps more hand-holding is required.

Clarification of how RWB events bring the observed air masses together was added in the form of the new figures 5 and 9 in the new version of the manuscript. The explicit analysis of backward trajectories allows one to determine since when a compact group of air parcels has been advected as such.

Line 201: Could/should the word "subsequently" be inserted right before "transported"(final word in this line). If that's not correct, then that means I haven't understood the discussion properly, and perhaps some clearer discussion is needed.

The word "subsequently" was inserted.

Figure 6: I'm afraid I found this figure hard to interpret/visualize. Perhaps, rather than multiple filled contours oriented vertically, might it be better to have just a few colored contour lines slicing horizontally. (e.g., Figure 6 doi:10.1002/2015JD023488).

Figure 6 was replaced with a new, hopefully clearer figure.

> Figure 9: Panel (a) is not discussed in the text so far as I could see. Also, panels (b)and (c) are discussed in reverse order. Consider discussing (a) and swapping the order of the panels so that they discussion and figure orders agree.

Figure 9 (now Figure 10 in the new version of the manuscript) has some additional material and the corresponding discussion in the text has been updated as well. All panels are now introduced.

> Line 401: "Large value+s+ of this term...". I think you should make it clear that (a) "this term" is J(x), correct, and also that (b) you mean large values after the iteration has converged, right?

We agree, Line 401 was corrected as suggested.